evolution

ageing, dietary restriction, longevity, reproduction, senescence

**Author for correspondence:**
Edward R. Ivimey-Cook
e-mail: e.ivimey-cook@uea.ac.uk

# Transgenerational fitness effects of lifespan extension by dietary restriction in *Caenorhabditis elegans*

Edward R. Ivimey-Cook, Kris Sales, Hanne Carlsson, Simone Immler, Tracey Chapman and Alexei A. Maklakov

School of Biological Sciences, University of East Anglia, Norwich Research Park, Norwich NR4 7TU, UK

  ERI-C, 0000-0003-4910-0443; KS, 0000-0002-7568-2507; SI, 0000-0003-1234-935X; TC, 0000-0002-2401-8120; AAM, 0000-0002-5809-1203

Dietary restriction (DR) increases lifespan in a broad variety of organisms and improves health in humans. However, long-term transgenerational consequences of dietary interventions are poorly understood. Here, we investigated the effect of DR by temporary fasting (TF) on mortality risk, age-specific reproduction and fitness across three generations of descendants in *Caenorhabditis elegans*. We show that while TF robustly reduces mortality risk and improves late-life reproduction of the individuals subject to TF ($P_0$), it has a wide range of both positive and negative effects on their descendants ($F_1$–$F_3$). Remarkably, great-grandparental exposure to TF in early life reduces fitness and increases mortality risk of $F_3$ descendants to such an extent that TF no longer promotes a lifespan extension. These findings reveal that transgenerational trade-offs accompany the instant benefits of DR, underscoring the need to consider fitness of future generations in pursuit of healthy ageing.

## 1. Introduction

Dietary restriction (DR), a reduction in nutrient intake without malnutrition, is an environmental intervention that robustly extends lifespan and/or improves health across a broad cross-taxonomic variety of organisms from yeast to mice to primates [1–5]. However, DR commonly reduces reproduction and long-term DR can be difficult to sustain in humans [6,7]. These considerations led to the development of alternative approaches such as DR mimetics and less demanding DR regimes such as different forms of temporary fasting (TF) [8,9]. TF, either for a distinct period of time or intermittently (i.e. repeatedly switching between periods of fasting or full-feeding), has been shown to increase lifespan in model organisms [10–14] and improve health in humans [15], and some forms of TF are currently being investigated for their potential to speed up patient recovery after surgery and chemotherapy [16]. Nevertheless, we know very little about the potential effects of DR, in whichever form it is implemented, on the fitness of offspring, and even less so about the transgenerational effects of DR on the fitness of more distant descendants.

Despite the positive effects of DR, several studies show that it can also be costly. For example, a recent study in *Drosophila melanogaster* fruit flies reported reduced survival and fecundity in individuals returned to a standard diet after a period of fasting, suggesting a hidden cost of improved survival under DR [17]. However, the data from model organisms are currently inconclusive, because other studies in *D. melanogaster* did not report such effects [18,19], while a study in *Caenorhabditis remanei* nematodes suggests that DR improves fitness upon return to normal feeding conditions [20]. Nevertheless, there are good reasons to believe that parental DR may affect offspring health and lifespan, but these effects will likely depend on the environmental conditions encountered by the offspring. Anticipatory parental effects may improve

offspring performance when offspring themselves are raised under DR [21] but may result in reduced fitness when offspring are raised in a standard environment [22,23]. The detrimental effects of such environmental mismatches between parents and their offspring have been shown previously [20,24] but are rarely investigated as a potential fitness cost of DR-mediated lifespan extension.

Recently, there has been a surge of interest in transgenerational effects where parental condition affects the health and fitness of not only offspring and grand-offspring but also of more distant descendants [25]. Such effects, if common, may have a profound influence on major evolutionary processes [25,26] and will probably have important implications for translational research [27]. Specifically, transgenerational trade-offs between parental fitness and fitness of distant descendants may constitute an obstacle for research programmes aimed at harnessing the power of DR for life- and health-span extension [28]. Alternatively, transgenerational transfer of the desired phenotype may be seen as an additional benefit. Some of the most spectacular examples of transgenerational effects come from the recent work on *C. elegans* nematodes [29–35]. Moreover, research suggests that larval starvation (the complete removal of food, which often results in malnutrition) in ancestors results in transgenerational inheritance leading to increased lifespan of $F_3$ offspring [29]. Thus, the combination of novel theoretical considerations and recent empirical discoveries calls for the investigation of transgenerational effects of ancestral DR on lifespan and fitness of descendants.

There are two main evolutionary models that explain the life-extending effect of DR, despite the overall reduction in resources. The resource allocation model suggests that animals experiencing food shortage will temporarily switch their metabolism from reproduction to somatic maintenance to increase their chances of survival until resources will become plentiful again [36]. An important extension of this model is the consideration of direct negative effects of reproduction on survival. Thus, by reducing reproduction in favour of survival, the animals both have relatively more resources for somatic maintenance and repair and suffer less damage from reproduction [36]. However, a recent model suggested that DR-mediated lifespan extension may be an unselected by-product of increased autophagy with the goal of maximizing reproduction using internal resources under the conditions when external resources are limited [37,38]. Interestingly, recent studies in fruit flies and nematodes suggest that the DR response is under neuronal control and can be manipulated by providing animals with food odour alone [39–42]. Suggesting that simply the presence of odour from food is sufficient to switch animal physiology from a self-preservation mode to a reproduction mode, reducing the benefits of DR. This technique provides us with an elegant research tool to test whether animals supress or maximize their reproduction under DR, but so far there have been no studies of the transgenerational fitness consequences of such treatments.

Here, we focused on addressing the following unresolved questions. (i) How does DR by TF affect mortality risk and reproductive ageing once the animals return to their standard food regime? (ii) How do offspring of TF parents perform in matching and mis-matching environments? (iii) Do transgenerational effects of ancestral fasting shape mortality risk and reproductive ageing of more distant descendants? (iv) Does reduced reproduction under DR represent a

response mediated by odour perception (or absence thereof)? We use *C. elegans* nematode worms, which are an established model for the study of both DR and transgenerational effects, to investigate how 2-day bacterial deprivation in early adulthood affects mortality risk and age-specific reproduction in ancestors and their descendants over a total of four generations. Our results reveal strong and previously unknown transgenerational costs of DR and demonstrate that *C. elegans* supress their own reproduction when nutrient limited. We use the results of this study to make major inferences about the adaptive nature of the DR response and suggest that the transgenerational costs of DR can be sufficiently severe to be considered in any applied programme aimed at lifespan extension via reduced nutrient intake.

## 2. Material and methods

### (a) Strains
*Caenorhabditis elegans* nematodes of the Bristol N2 wild-type strain from the *Caenorhabditis* Genetics Centre were used in all assays. Populations were thawed from −80°C and underwent bleaching (exposure to a sodium hypochlorite, NaOCl and sodium hydroxide, NaOH, solution) and egg-lays prior to the start of the experiment to synchronize the developmental age of the $P_0$ individuals. Populations were then kept in climate chambers set to 20°C, 60% RH and continual darkness. Worms were maintained on standard Nematode Growth Medium (NGM) agar in Petri dishes. These were 90 mm wide for population maintenance and 35 mm for individual culture. The NGM agar contained a fungicide (100 µg ml$^{-1}$ nystatin) and antibiotics (100 µg ml$^{-1}$ ampicillin and 100 µg ml$^{-1}$ streptomycin) to prevent infection. In all cases, nematodes were fed using antibiotic resistant *Escherichia coli* OP50-1 (pUC4 K), from J. Ewbank at the Centre d'Immunologie de Marseille-Luminy, France. The OP50-1 (pUC4 K) *E. coli* were thawed from −80°C, streaked on LB plates containing ampicillin (100 µg ml$^{-1}$) and streptomycin (100 µg ml$^{-1}$), incubated at 37°C for 16 ± 1 h and kept at 6°C. A single bacterial colony was inoculated per 40 ml of LB broth containing the same antibiotic concentrations then incubated at 37°C for 16 ± 1 h. The *E. coli* solution was then pipetted onto NGM agar Petri dishes and incubated at 20°C for 18 ± 6 h to grow ad libitum bacterial lawns prior to use. The time from bacterial thawing to use was within a month. NGM agar plates were used within a few weeks (max = 4 weeks) of preparation.

### (b) Experimental set-up
The $P_0$ generation consisted of four distinct treatments (figure 1a): (i) ad libitum (AL) control plates consisted of standard NGM agar plates topped with an *E. coli* lawn. (ii) TF plates contained no peptone (an ingredient necessary for bacterial growth) within the NGM agar and were not seeded with *E. coli*, which, together, reduced the potential growth of an alternate food source while the worms were fasting. (iii) Food odour (TF(FO)) plates were designed so worms could detect the presence of an *E. coli* food source but were unable to access it. They consisted of a base NGM agar layer seeded with *E. coli* then incubated overnight. A second NGM agar layer, without peptone and without bacterial seeding, was then added. (iv) The fourth treatment (AL(FO)) was a positive control plate, combining the food odour treatment with ad libitum, by layering a TF(FO) plate with a top agar layer containing peptone and bacterial seeding. The time allowed for bacterial growth between treatments was standardized. Two days prior to use, the bottom layers for the TF(FO) and AL(FO) positive control treatments were incubated. All

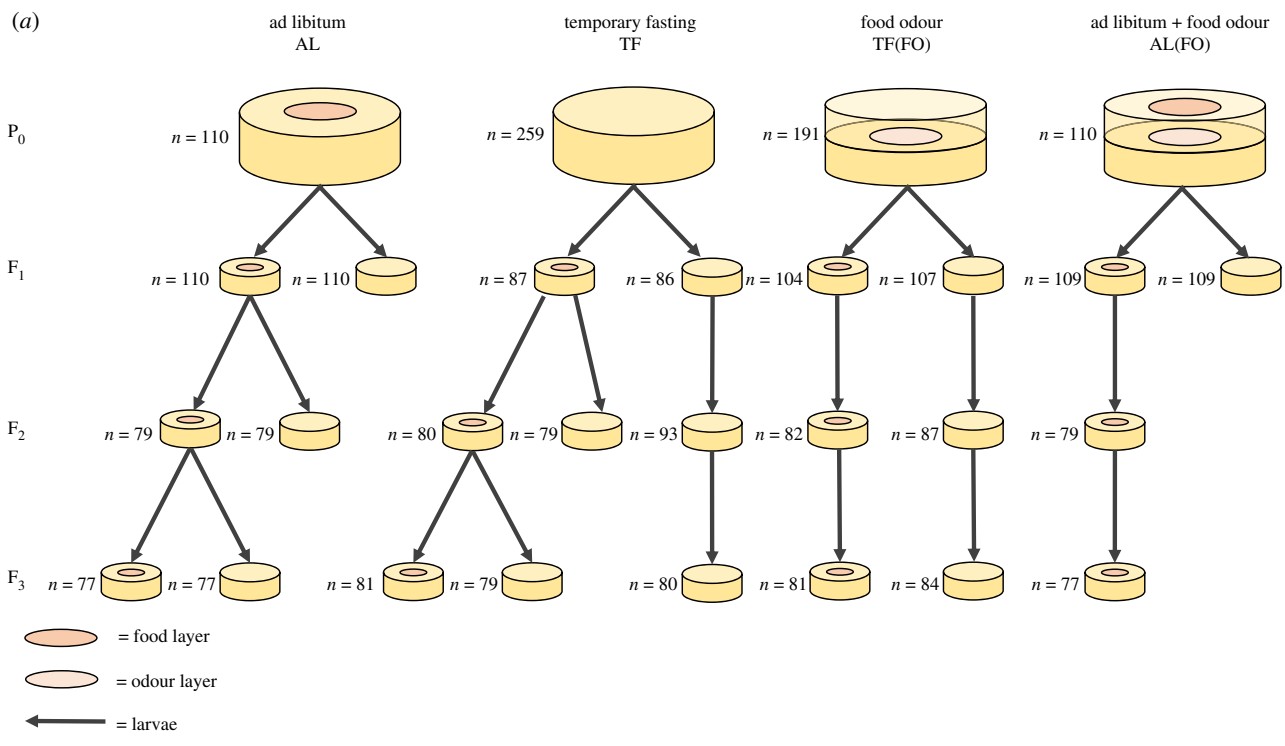

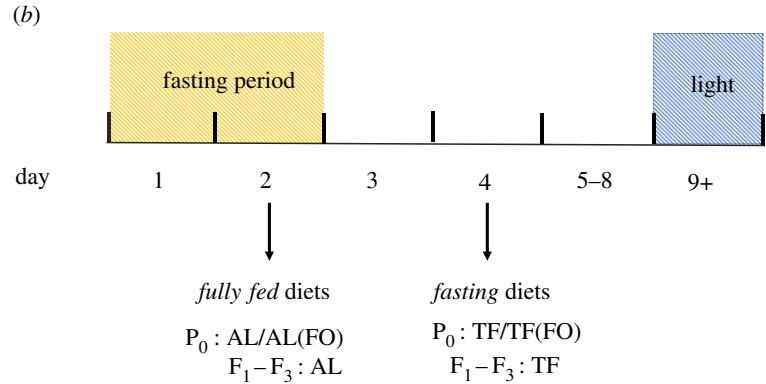

**Figure 1.** (a) A visual representation of the dietary treatments and lineages studied (from $P_0$ to $F_3$) with corresponding sample sizes across three (for $P_0$ and $F_1$) or two (for $F_2$ and $F_3$) experimental blocks. Larvae were transferred (shown with black arrows) and placed individually onto either ad libitum (AL) or temporary fasting (TF), denoted with or without the presence of a food layer. (b) Graphical representation of the dietary paradigm used for $P_0$–$F_3$ individuals. Day of larval collection is denoted with the black arrows. (Online version in colour.)

plates were incubated for 1 day to grow the upper seeding and to standardize incubation time across all four treatments.

## (c) Transgenerational lifespan and fitness assays

To study the transgenerational effects of $P_0$ diet treatments, individual late-$L_4$ stage nematodes were randomly exposed to one of four dietary treatments (figure 1a; sample sizes varied for each treatment owing to disproportionate day 1 and two mortality of TF individuals by walling ($n = 110$ for AL/AL(FO), $n = 191$ for FO, $n = 259$ for TF). They remained exposed to these dietary conditions for 2 days prior to transfer onto standard ad libitum control plates seeded with *E. coli*. The presence or absence of food led to corresponding shifts in reproductive schedule (see Results). Thus, to generate sufficient progeny for future generations ($F_1$–$F_3$), we took offspring from different days for each of the four dietary treatments (day 2 for AL/AL(FO) and day 4 for TF/TF(FO); figure 1b). The different choice of days allowed us to obtain sufficient offspring from all treatments and ensure that larvae were raised in fully fed environments across

all treatments. To generate successive generations, eggs were allowed to hatch and develop for 2 days upon which two surviving larvae from each parent were randomly allocated into either AL or TF conditions (in some cases, this number was increased to compensate for higher than predicted loss of individuals). This continued for two further generations (until $F_3$; figure 1a). To ensure a sufficiently large sample size, the lack of a priori expectations, and the need to focus our tests around the specific questions above, certain lineages were only continued until $F_1$ (figure 1a).

Every generation was assayed for both lifespan and fitness by transferring onto new plates every 24 h up and kept in climate chambers set to 20°C, 60% RH with continual darkness. After day 8 of adulthood, all surviving worms were then haphazardly positioned on the top of a clear plastic tray and placed on a shelf at random within an incubator at 20°C with a 8 h : 16 h L : D cycle at approximately 7000 Lux and transferred every 2 days onto new plates. Eggs laid within the 8-day reproductive period were allowed to hatch and develop for 2 days before hatched larvae were killed at 42°C for 3.5 h and subsequently counted.

For lifespan, death was defined as the absence of worm movement in response to touch. A worm was censored if the time of death was unknown, if it went missing, or if the worm gained access to the layer of food in the middle of the plate. These assays were repeated over separate blocks (three for $P_0$ and $F_1$ and two for $F_2$ and $F_3$). The experimental design of the first block differed slightly, with individual worms (from $P_0$ to $F_1$) placed in light from day 1 of reproduction, although the larvae still developed in darkness. As all treatments were otherwise treated similarly, with or without light from day 1 of reproduction, we kept these block 1 data within the analysis (see electronic supplementary material, figures S1A–D and S2A–D).

## (d) Worm length and egg area assay

To quantify the effects of dietary regime on important aspects of worm development and growth across multiple generations, we measured $P_0$ and $F_1$ worms at multiple time points throughout their lifespan ($P_0$ = days 2 and 4 of adulthood; $F_1$ = egg, $L_1$ and $L_4$). This was done in a separate assay to above; however, while the dietary regimes and plate preparation remained the same, the number of worms per small agar plate was increased to 5. Importantly, the $L_4$ stage of the $F_1$ generation was split into two time periods, chronological (the standardized time that all treatments were set up during the experimental assay) and biological (the approximate biological time that worms on each respective plate became late-$L_4$). This particular life-stage was separated to observe the effects of dietary regime on initial worm size at experimental set-up, and, as DR often leads to an increase in development time, to see if this size was prolonged until sexual maturation. All photos were taken using the Leica Application Suite software v. 4.13 and measurements of worm length or egg area were calculated using the measuRe package v. 0.0.0.900 (github.com/joelpick/measuRe) in R v. 4.0.3 [43]. Two measurements for each worm or egg were taken and an average value was created and used in subsequent visualization with estimation plots from the dabestR package v. 0.3.0 [44].

## (e) Statistical analysis

All analyses were performed using R v. 4.0.3 [43].

Data used for the survival and reproduction analyses differed slightly. While both datasets contained data from individuals where the time of death was known, individuals that died due to matricide were omitted from the survival analysis. For reproduction, these individuals were included but censored individuals (i.e. those where the exact time of death was not known) were omitted from the LRS and $\lambda ind$ analyses.

For all models involving $P_0$, a generation-level factor of 'treatment' was added as a fixed effect with a random effect of 'founder' to account for possible pseudo-replication. As there were only three levels of experimental block, an additional fixed effect of 'block' was added. For $F_1$, to explicitly model the relationship between parent–offspring environment matching in F1 (Question 2), models were run which comprised 'parent treatment' and 'treatment' and the higher-order interaction between the two (see electronic supplementary material, table S18). For models involving $F_2$–$F_3$, we fitted a new fixed factor of dietary 'lineage' (instead of 'treatment') to compare between different levels of dietary history (in addition to the other fixed factor of 'block'). We also added an additional random effect of 'parent ID' instead of 'founder'. For reproduction in the $F_2$–$F_3$ generation, data were subsetted into the two dietary environments (TF and AL) to more clearly determine the effects of great-grand-/ grand-/parental diets acting on offspring fitness.

For the most part, our survival data adhered to the assumption of proportional hazards required to by the Cox proportional hazard models. However, data involving survival in the $F_3$ generation did not conform to this assumption (GLOBAL term from cox.zph test $\chi^2$ = 29.7, $p$ = 0.030). Therefore, for consistency, we instead used an event history analysis (see [45]) using the glmmTMB package v. 1.0.2.900 [46,47]. Qualitatively, this is similar to a Cox proportional hazard model, but presents results in terms of probability of death per day (or mortality risk) and is modelled using a binomial distribution. Individuals are scored daily throughout their life with 0 denoting alive and 1 denoting dead (N.B. censoring is achieved by replacing this terminal 1 with a 0). As a result of repeatedly measuring the same individual, we added an additional random effect of 'worm ID' nested within the higher-order random effect (see above and electronic supplementary material, table S18). In addition, a random effect of 'day' was added to account for variation between each time interval. The coefficients from these models were then visualized in a forest plot created using ggplot2 v. 3.3.3 [48].

For reproduction, three measures of fitness were analysed. Typically, age-specific reproduction within C. *elegans* nematodes are overdispersed with significant evidence of zero-inflation, particularly when individuals are dietary restricted. If these measures were identified as zero-inflated or overdispersed (by simulating the residuals and testing for zero-inflation in two distinct models, a Poisson model and a Poisson model with an observation-level random effect; using the DHARMa package v. 0.3.3, see [49]), an additional zero-inflation/dispersion component, and a variety of error distribution were fitted using the glmmTMB package. We fitted similar covariates as the survival models; however, we added both the quadratic and linear fixed effects of 'day' and their interaction with either 'treatment' (for $P_0$) or 'lineage' (for $F_1$–$F_3$) to capture the change in reproduction for each dietary lineage over time. In addition, we also added a random effect of 'worm ID' to account for repeatedly measuring the same individual. Age-specific reproductive curves were then visualized using ggplot2.

The second measure of fitness was the total number of offspring produced by each individual, or lifetime reproductive success (LRS). Similar fixed and random effects were used as in the mixed-effects survival model. Differences between treatments were also analysed by bootstrapping and displayed in estimation plots from the dabestR package.

The last measure of fitness was individual fitness ($\lambda ind$) and was obtained by constructing individual-based age-structured matrices (Leslie matrices [50]) and calculating the dominant eigenvalue using the lambda function from the popbio package v. 2.7 [51]. Two days of development time were added onto the fertility schedule and represented time from egg to adulthood. These values were then analysed using glmmTMB with a Gaussian error structure and with similar factors as the previous models. Similarly, individual fitness was visualized on bootstrapped estimation plots from the dabestR package.

Worm length and egg area were both analysed using glmmTMB with a Gaussian error structure. Each model had 'treatment' as a fixed effect with 'worm ID' nested within 'plate' as random effects to account for repeatedly measuring the same worm twice and for pseudo-replication of worms from the same agar plate.

In all cases, aside from the individual fitness and survival model, model selection was performed to identify the best fitting error distribution, zero-inflation and dispersion parameters for each response variable; chosen as the model with the lowest Akaike information criterion (AIC), and with acceptable levels of dispersion and zero-inflation (identified using DHARMa; see electronic supplementary material for model selection tables). Lastly, type three Wald tests were performed to determine the overall effect of 'dietary treatment' or 'lineage' and subsequent higher order interactions on the various measured traits using the Anova function from the car package v. 3.0-10 [52]. For the ease of reading, effect sizes from each of the reproduction models were reported as back-transformed estimated marginal means using the emmeans package v. 1.5.4 [53].

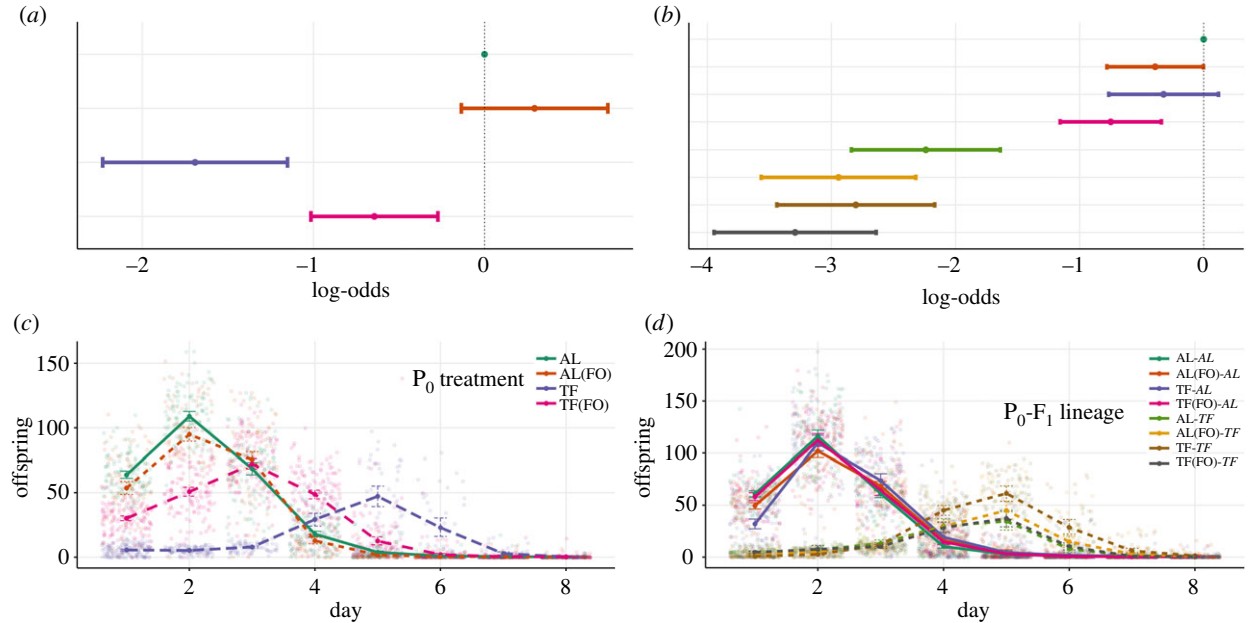

**Figure 2.** Plots of (*a,b*) mortality and (*c,d*) reproduction for (*a,c*) $P_0$ and (*b,d*) $F_1$. Colours represent the various dietary treatments or lineages for each generation, while line-type (for *d*) represent either ad libitum (AL, solid line) or temporary fasting (TF, dotted line) environment. (*a,b*) Points represent coefficients from a mixed effects event history model with 95% confidence intervals with either AL (*a*) or AL−AL (*b*) set to 0 as the reference. (*c,d*) Points represent mean values with 95% confidence intervals. N.B. (FO) represents the presence of a food odour layer. (Online version in colour.)

## 3. Results

### (a) $P_0$ mortality

The mortality of $P_0$ individuals was significantly impacted by their dietary treatment ($\chi^2 = 53.94$, $p < 0.001$). Individuals that were temporarily fasted (TF) exhibited a significant reduction in mortality, in comparison to ad libitum (AL) individuals (*Log-odds* = −1.69, 95%: −2.23, −1.15, $p < 0.001$; figure 2*a*; electronic supplementary material, table S1A,B, figure S4A). In addition, while still reduced in comparison to the mortality rates of AL and AL(FO) (*Log-odds* = −0.64, 95%: −1.01, −0.27, $p < 0.001$; *Log-odds* = −0.94, 95%: −1.35, −0.52, $p < 0.001$), individuals that were simply given the odour of food (TF(FO)) had mortality rates significantly greater than temporarily fasted individuals (*Log-odds* = 1.05, 95%: 0.55, 1.54, $p < 0.001$; figure 2*a*, electronic supplementary material, table S1A,B, figure S4A).

### (b) $P_0$ reproduction

In a similar manner to mortality, all reproductive measures were significantly affected by dietary treatment (LRS: $\chi^2 = 189.10$, $p < 0.001$; $\lambda$ind: $\chi^2 = 2210.79$, $p < 0.001$) and its interaction with both the linear and quadratic effects of age, respectively ($\chi^2 = 38.16$, $p < 0.001$; $\chi^2 = 185.52$, $p < 0.001$). TF individuals exhibited a much delayed and reduced reproductive schedule in comparison to the TF(FO) treatment, which had an intermediate effect (figure 2*c*). In particular, these TF(FO) individuals exhibited a delayed reproductive peak in comparison to the AL(FO) and AL treatments, but a far advanced peak when compared with TF individuals, who peaked on Day 5 (figure 2*c*). Moreover, TF(FO) individuals had far greater LRS in comparison to the TF treatment (mean LRS: TF = 56.1, TF(FO) = 211.0; ratio = 3.76, 95%: 2.86, 4.96, $p < 0.001$; figure 3*a*; electronic supplementary material, table S3A–D). This earlier peak in reproduction also resulted in higher $\lambda$ind for the TF(FO) treatment relative to TF individuals (mean $\lambda$ind: TF = 2.07, TF(FO) = 3.60; difference = 1.54,

95%: 1.43, 1.65, $p < 0.001$; figure 3*c*; electronic supplementary material, table S4A,B).

### (c) $F_1$ mortality

N.B. the order of the dietary lineage shown indicates **$P_0$**–$F_1$.

Mortality in the $F_1$ generation was significantly affected by individual treatment ($\chi^2 = 151.29$, $p < 0.001$), with TF individuals exhibiting reduced mortality risk in comparison to the AL treatment (TF: *Log-odds* = −2.45, 95%: −2.84, −2.06, $p < 0.001$, figure 2*b*; electronic supplementary material, table S5A,B, figure S4B). Two generations of fasting (**TF**–TF) produced no detectable effects on mortality in comparison to individuals with different parental treatments (AL–TF: *Log-odds* = 0.56, 95%: −0.13, 1.26, $p = 0.113$; **TF(FO)**–TF: *Log-odds* = −0.49, 95%: −1.15, 0.17, $p = 0.149$; **AL(FO)**–TF: *Log-odds* = −0.14, 95%: −0.81, 0.53, $p = 0.685$; figure 2*b*; electronic supplementary material, table S5A,B, figure S4B). By contrast, mis-matched parent–offspring environments for **AL**–TF individuals resulted in increased mortality when compared with offspring from AL(FO) and TF(FO) parents (**AL(FO)**–TF: *Log-odds* = 0.70, 95%: 0.03, 1.38, $p = 0.041$; **TF(FO)**–TF: *Log-odds* = 1.05, 95%: 0.36, 1.75, $p = 0.003$; figure 2*b*; electronic supplementary material, table S5A,B, figure S4B). However, in contrast, both the offspring from TF(FO) and AL(FO) parents exhibited decreased mortality when placed in AL conditions (**TF(FO)**–AL: *Log-odds* = −0.75, 95%: −1.15, −0.34, $p < 0.001$; **AL(FO)**–AL: *Log-odds* = −0.39, 95%: −0.78, −0.00, $p = 0.049$; figure 2*b*; electronic supplementary material, table S5A and B and figure S4B). Overall, the interaction between parent–offspring environment remained statistically non-significant ($\chi^2 = 0.819$, $p = 0.845$).

### (d) $F_1$ reproduction

The influence of parental diet acting on offspring reproduction and fitness depended significantly on an individual's dietary treatment (LRS: $\chi^2 = 22.01$, $p < 0.001$; $\lambda$ind: $\chi^2 = 27.99$, $p < 0.001$)

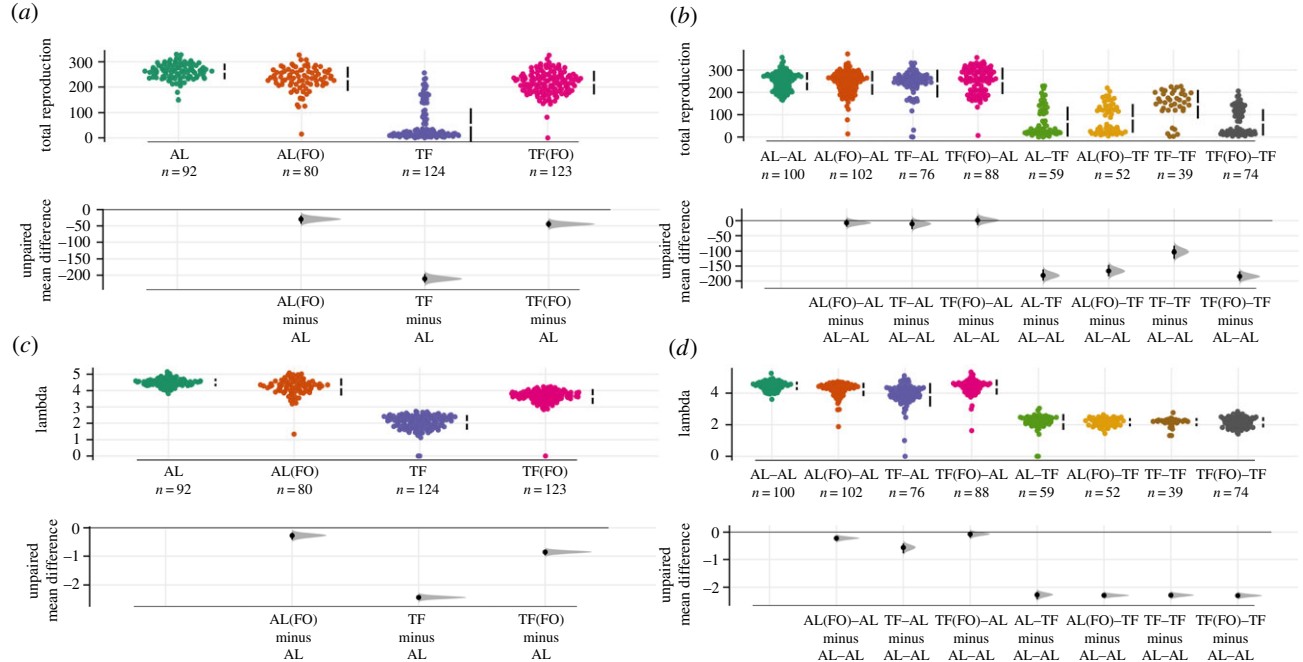

**Figure 3.** Plots of (*a,b*) LRS and (*c,d*) individual fitness (λ*ind*) for (*a,c*) P$_0$ and (*b,d*) F$_1$. Colours represent differing dietary treatments or lineages for each generation. In each case, the top panels represent raw data with the black bar showing the mean and 95% confidence intervals. Below each group of raw data, corresponding sample sizes are given with *n*. Bottom panels represent estimation plots with bootstrapped mean differences and confidence intervals between the reference (AL for P$_0$ and AL–AL for F$_1$) and other treatments/lineages. N.B. (FO) represents the presence of a food odour layer. (Online version in colour.)

and the interaction with both forms of age (linear: $\chi^2 = 73.25$, $p < 0.001$; quadratic: $\chi^2 = 54.38$, $p < 0.001$). For individuals in the AL conditions, both the LRS and λ*ind* were reduced in offspring from **TF**–AL parents in comparison to offspring from **AL**–AL parents (mean LRS: **AL**–AL = 241.5, **TF**–AL = 216.2; ratio = 1.12, 95%: 1.03, 1.21, $p = 0.009$; mean λ*ind*: **TF**–AL = 3.79, **AL**–AL = 4.44; difference = 0.64, 95%: 0.51, 0.77, $p < 0.001$; figure 3*b*, *d*; electronic supplementary material, tables S7A–D and S8A and B). The opposite pattern was shown for individuals placed under TF. LRS was positively affected by parental treatment, with **TF** parents producing offspring with higher total reproductive count than offspring from **AL** parents (mean LRS: **TF**–TF = 129.6, **AL**–TF = 69.7, ratio = 1.86, 95%: 1.27, 2.7, $p = 0.001$; figure 3*b*; electronic supplementary material, tables S7A–D, S8A,B). However, as the reproductive peak was on day 5 rather than on days 1 and 2, a significant decrease was found in individual fitness (mean λ*ind*: **TF**–TF = 2.00, **AL**–TF = 2.19; difference = −0.18, 95%: −0.35, −0.01, $p = 0.034$; figure 3*d*; electronic supplementary material, tables S7A–D, S8A,B). Interestingly, offspring from TF(FO) parents exhibited similar patterns as offspring from AL parents, with reduced LRS (mean LRS: **TF(FO)**–TF = 65.0, ratio = 2.00, 95% 1.40, 2.85, $p < 0.001$) and increased λ*ind* in TF (although not statistically significant, the direction of effect remained the same; mean λ*ind*: **TF(FO)**–TF = 2.16; difference = −0.16, 95%: −0.32, −0.007, $p = 0.061$), and no detectable differences in AL (mean LRS: **TF(FO)**–AL = 240.5, ratio = 1.00, 95%: 0.95, 1.06, $p = 0.874$; mean λ*ind*: **TF(FO)**–AL = 4.36; difference = 0.08, 95%: −0.04, −0.19, $p = 0.199$; electronic supplementary material, tables S7A–D, S8A,B).

### (e) F$_2$ mortality
N.B. the order of the dietary lineage shown indicates **P$_0$**–F$_1$–F$_2$.

Mortality was significantly impacted by grandparental dietary treatment. Grandparents that were placed within TF or TF(FO) conditions produced grand-offspring that exhibited a significant *increase* in mortality when placed

within AL conditions (**TF**–AL–AL: *Log-odds* = 1.39, 95%: 0.87, 1.92, $p < 0.001$; **TF(FO)**–AL–AL: *Log-odds* = 1.07, 95%: 0.57, 1.57, $p < 0.001$; figure 4*a*; electronic supplementary material, table S9A,B, figure S4C). As expected, TF again promoted a reduction in mortality for all treatments (electronic supplementary material, table S9A,B). However, the degree to which lifespan was increased was based on the number of successive generations of ancestral fasting (figure 4*a*). Those with two or three generations of cumulative fasting (**TF**–TF–TF and **TF(FO)**–TF–TF) exhibited a reduced lifespan extension in comparison to those with only one generation (**AL**–AL–TF) (**TF**–TF–TF: *Log-odds* = −2.09, 95%: −2.95, −1.22, $p < 0.001$; **TF(FO)**–TF–TF: *Log-odds* = −1.98, 95%: −2.90, −1.07, $p < 0.001$; figure 4*a*; electronic supplementary material, table S9A,B, figure S4C).

### (f) F$_2$ reproduction
Grandparental diet had no effect on any of the measured reproductive traits when individuals were placed in AL conditions. However, the mean values suggested that both TF and TF(FO) grandparents produced grand-offspring with increased average LRS in comparison to **AL**–AL–AL (mean LRS: **AL**–AL–AL = 268.00, **TF**–AL–AL = 278.00 and **TF(FO)**–AL–AL = 279.00); however, these differences were not statistically significant (ratio = 0.967, 95%: 0.92, 1.01, $p = 0.161$; ratio = 0.961, 95%: 0.92, 1.01, $p = 0.09$; figure 5*a*; electronic supplementary material, table S11A–C) and furthermore resulted in no difference between λ*ind* values (mean λ*ind*: **AL**–AL–AL = 4.46, **TF**–AL–AL = 4.42 and **TF(FO)**–AL–AL = 4.43; figure 5*c*; electronic supplementary material, table S12A). In the TF conditions, TF grandparents produced grand-offspring with significantly higher LRS in comparison to individuals produced from AL grandparents (mean LRS: **TF**–TF–TF = 132.8, **AL**–AL–TF = 98.5, ratio = 1.35, 95%: 1.02, 1.79, $p = 0.038$; figure 5*a*; electronic supplementary material, table S11D–F). Lastly, within the TF environment there was

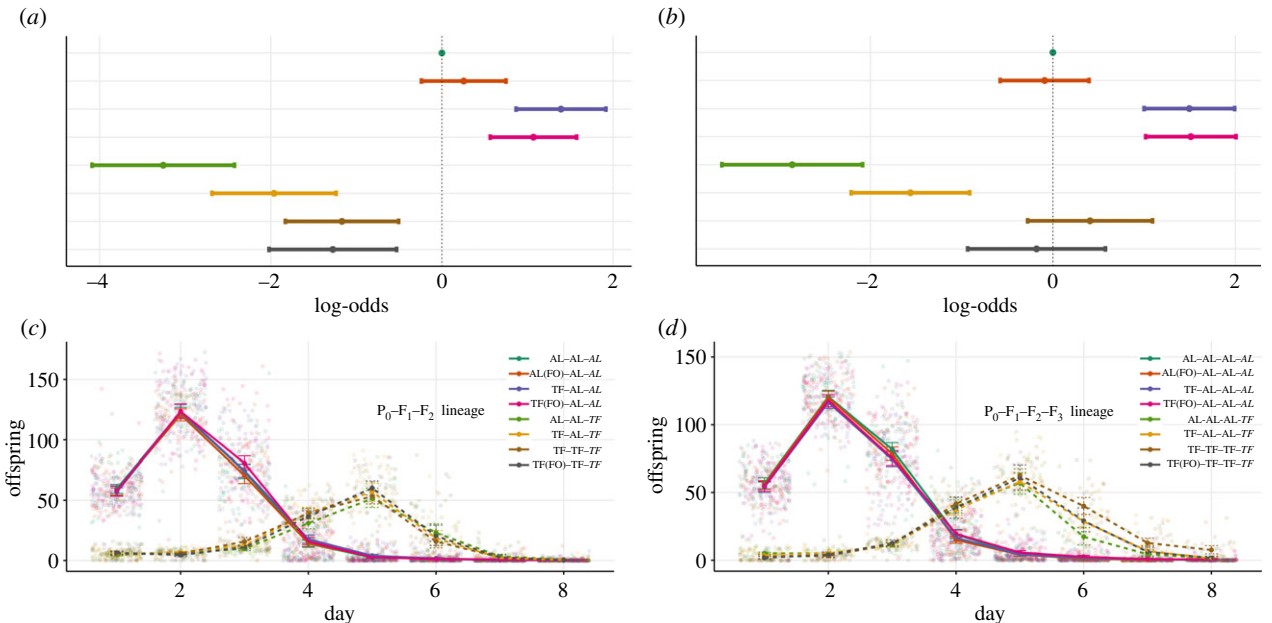

**Figure 4.** Plots of (*a*,*b*) mortality and (*c*,*d*) reproduction for (*a*,*c*) $F_2$ and (*b*,*d*) $F_3$. Colours represent the various dietary lineages for each generation, whilst line-type represent either ad libitum (AL, solid line) or temporary fasting (TF, dotted line) environment. (*a*,*b*) Points represent coefficients from a mixed effects event history model with 95% confidence intervals with either (*a*) AL-AL-AL or (*b*) AL−AL−AL−AL set to 0 as the reference. (*c*,*d*) Points represent mean values with 95% confidence intervals. N.B. (FO) represents the presence of a food odour layer.

a detectable decrease in $\lambda ind$ for offspring born from AL grandparents (mean $\lambda ind$: **TF**–TF–TF = 2.48, **AL**–AL–TF = 2.29; difference = −0.189, 95%: −0.31, −0.07, $p = 0.002$; figure 5*c*; electronic supplementary material, table S12B).

## (g) $F_3$ mortality

N.B. the order of the dietary lineage shown indicates $\mathbf{P_0}$–$F_1$–$F_2$–$F_3$.

In a similar manner to $F_2$, the mortality risk of individuals was negatively impacted by great-grandparental treatment. With great-grandparents that were placed within TF or TF(FO) conditions producing great-grand-offspring with increased mortality risk in comparison to other treatments (**TF**–AL–AL–AL: *Log-odds* = 1.50, 95%: 1.00, 1.99, $p < 0.001$; **TF(FO)**–AL–AL–AL: *Log-odds* = 1.51, 95%: 1.02, 2.01, $p < 0.001$; figure 4*b*; electronic supplementary material, table S13A, figure S4D). Strikingly, the positive effects of TF, which prevailed throughout the previous generations, disappeared. With the cumulative effects of four successive generations (three for the TF(FO) great-grand-parental treatment) in TF environments causing a distinct lack of lifespan increase in comparison to four generations of AL feeding (**TF**–TF–TF–TF: *Log-odds* = 0.41, 95%: −0.28, 1.10, $p = 0.243$; **TF(FO)**–TF–TF–TF: *Log-odds* = −0.18, 95%: −0.93, 0.58, $p = 0.642$; figure 4*b*; electronic supplementary material, table S13A, figure S4D). Lastly, those individuals with only one successive generation of fasting still exhibited the observable decrease in mortality observed from the $P_0$ generation (**AL**–AL–AL–TF: *Log-odds* = −2.86, 95%: −3.63, −2.01, $p < 0.001$; **TF**–AL–AL–TF: *Log-odds* = −1.56, 95%: −2.20, −0.91, $p < 0.001$; figure 4*b*; electronic supplementary material, table S13A, figure S4D).

## (h) $F_3$ reproduction

Great-grandparental diet had detectable transgenerational effects acting on the LRS and $\lambda ind$ of individuals in both the AL and TF environments (electronic supplementary material, table S2Z–AH; figures 4*d* and 5*b*,*d*). Great-grand-offspring of TF individuals exhibited decreased LRS in AL environments compared to the AL control lineage (mean LRS: **AL**–AL–AL–AL = 280, **TF**–AL–AL–AL = 268, ratio = 1.04, 95%: 1.01, 1.08, $p = 0.02$; figure 5*b*; electronic supplementary material, table S15A,B). In addition, great-grand-offspring of AL individuals exhibited decreased LRS in TF environments compared to the cumulative TF treatment (mean LRS: **TF**–TF–TF–TF = 155, **AL**–AL–AL–TF = 113, ratio = 1.37, 95%: 1.03, 1.83, $p = 0.031$; figure 5*b*; electronic supplementary material, table S15C–E). For individual fitness, there were no detectable transgenerational effects when individuals were placed in AL environment; however, individuals exhibited decreased $\lambda ind$ values when placed within TF environments (electronic supplementary material, table S16A,B). In particular, individuals produced from three or more generations of cumulative fasting (including great-grand-offspring produced from four generations of successive fasting and from TF(FO) great-grandparents) exhibited significantly lowered fitness in comparison to those with only one generation of successive fasting (mean $\lambda ind$: **TF**–TF–TF–TF: 2.03, **AL**–AL–AL–TF: 2.34, **TF(FO)**–TF–TF–TF: 2.00, **TF**–AL–AL–TF: 2.35; **TF**–TF–TF–TF/**AL**–AL–AL–TF = difference: −0.308, 95%: −0.587, −0.029, $p = 0.031$; **TF**–TF–TF–TF/**TF**–AL–AL–TF = difference: −0.313, 95%: −0.603, −0.023, $p = 0.034$; figure 5*d*; electronic supplementary material, table S16A,B).

## (i) Worm development and growth

Diet significantly impacted several measures of worm development and growth across two generations. Namely, TF adult worms measured on both days 2 and 4 exhibited significantly reduced body length in comparison to AL adults (day 2 mean length (mm) = AL: 1.04, TF: 0.74; difference = −0.304, 95%: −0.328, −0.279, $p < 0.001$; day 4 mean length (mm) = AL: 1.14, TF: 0.90; difference = −0.238, 95%: −0.280, −0.196, $p < 0.001$; electronic supplementary material, figure S3A, table S17A,B). In addition, eggs produced from TF adults were smaller in area in comparison to those produced from

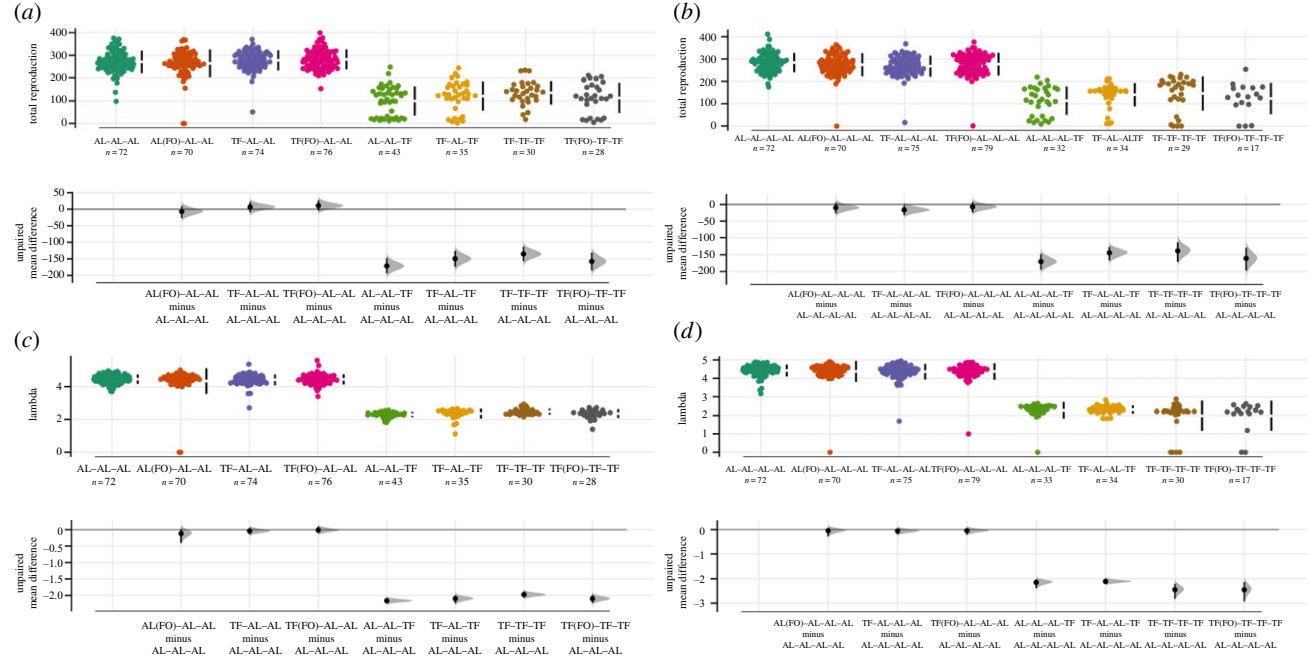

**Figure 5.** Plots of (*a,b*) LRS and (*c,d*) individual fitness ($\lambda ind$) for (*a,c*) $F_2$ and (*b,d*) $F_3$. Colours represent differing dietary lineages for each generation. In each case, the top panels represent raw data with the black bar showing the mean and 95% confidence intervals. Below each group of raw data, corresponding sample sizes are given with *n*. Bottom panels represent estimation plots with bootstrapped mean differences and confidence intervals between the reference (AL−AL−AL for $F_2$ and AL−AL−AL−AL for $F_3$) and other lineages. N.B. (FO) represents the presence of a food odour layer. (Online version in colour.)

AL parents (mean egg area (mm$^2$) = AL: 0.0014, TF: 0.0013; difference = −0.0001, 95%: −0.0002, −0.00004, $p = 0.001$; electronic supplementary material, figure S3B, table S17C). Furthermore, this resulting reduction in TF egg size led to a corresponding decrease in $L_1$ length ($L_1$ mean length (mm) = AL: 0.24, TF: 0.20; difference = −0.0321, 95%: −0.04, −0.023, $p < 0.001$; electronic supplementary material, figure S3C, table S14D).

When comparing $L_4$ length, two time periods were considered, the chronological and biological timing of late-L4 (see Material and methods). While these two periods generally overlapped for AL, AL(FO) and TF(FO) worms (with chronological time matching the biological timing of late-L4), there was a delay in the biological timing of late-$L_4$ TF worms. When comparing length at chronological time between treatments, TF worms exhibited significantly reduced body size in comparison to AL worms ($L_4$-chronological mean length (mm) = AL = 0.72, TF = 0.64, difference = −0.086, 95%: −0.12, −0.05, $p < 0.001$; electronic supplementary material, figure S3D, table S17E). Interestingly, when comparing length at the biological time point of late-$L_4$, TF worms no longer exhibited this reduction in length ($L_4$-biological mean length (mm) = AL = 0.72, TF = 0.71, difference = −0.01, 95%: −0.03, 0.01, $p = 0.274$; electronic supplementary material, figure S3E, table S17F).

## 4. Discussion

Most strikingly, we found evidence of several negative transgenerational effects of DR by TF in $P_0$ on the mortality and fitness of individuals from the $F_3$ generation. Specifically, great-grandparental DR increased mortality risk and reduced fitness of $F_3$ descendants. Contrary to previous work showing that larval starvation can produce positive transgenerational effects on lifespan in *C. elegans*, we demonstrate that TF in adulthood can result in detrimental transgenerational effects on lifespan by increasing mortality risk. We note, however, that this result

could in part be due to marked differences in dietary paradigm between both experiments. Here, we *fasted* individuals for 2 days during adulthood, whereas the latter experiment *starved* individuals during the first larval phase for 6 days. This distinction is important, as TF and other forms of DR are said to occur in the absence of malnutrition, whereas starvation implies the opposite. Moreover, individuals produced from three generations of TF no longer displayed the classical reduction in mortality risk associated with DR and exhibited significantly reduced individual fitness. Taken together, these results highlight previously unknown long-term costs of DR which may have significant detrimental effects that only manifest in distant generations.

Besides the clear inter- and transgenerational trade-offs associated with the DR response, we also found that olfactory cues influence mortality across several generations. $P_0$ individuals exposed to the odour of food but placed in the same environment as DR individuals exhibited increased reproduction at the cost of reduced lifespan extension. Moreover, $F_1$ offspring produced from food odour DR parents behaved in a similar manner to offspring born from ad libitum parents. Not only does this suggest that parental effects regarding dietary condition are reliant upon accurate environmental cues being passed to the next generation mediated through variable egg size, but also that investment into survival is largely in response to food-related odours which reduces longevity through the activation of insulin/IGF-1 pathways [40,54]. It is possible that nematodes are reluctant to lay eggs in the environment perceived as devoid of food and that reduced reproduction under DR is driven partly by the lack of resources to produce gametes and also as a result of an adaptive reproductive strategy by the organism. Taken together, these results are largely in line with the resource reallocation hypothesis [36] where individuals under DR reallocate resources from reproduction into somatic maintenance to increase the probability of survival until the next reproductive opportunity.

Our results answer a number of important questions regarding lifespan extension via DR in parents. Consistent with previous research [2,20,42,55], DR via TF resulted in a detectable reduction in mortality risk at the cost of reduced reproduction in the $P_0$ generation. Moreover, we found when individuals were placed back onto a standard food regime post-DR (day 3), reproduction steadily increased until a peak occurred on day 5. This contrasts with the findings of Mccracken et al. [17] who found that the survival and fertility of D. melanogaster decreased immediately following a return to a rich diet after a period of DR; however, they note that integral to this decrease in fitness is the relative duration of DR prior to rich feeding. It is possible, therefore, that we may have seen similar patterns of mortality exacerbation if nematodes remained within this TF environment for longer.

Our results also show that DR in $P_0$ not only affects the present generation but also has long-term effects for up to three subsequent generations. We identified significant long-term effects of both parental and ancestral diet manifesting in changes to survival and fitness of subsequent generations. In fact, if the parental generation and $F_1$ offspring were exposed to the same dietary treatment, the benefits carried on across both generations. This in part may be mediated through delayed development time allowing $F_1$ progeny to better match reproduction on return to an AL environment post-TF. This finding suggests the transmission of information allows offspring to anticipate potentially adverse conditions, such as temporary food shortage, resulting in increased performance. Although different in methodology, these results are largely consistent with intergenerational phenotypic plasticity acting between mother and offspring identified by Hibshman et al. [21]. However, we extend this work by showing that the offspring of DR parents pay a price of reduced fitness when raised in standard environmental conditions. Thus, the parental DR response can be costly for the offspring and its adaptive nature depends on whether the current environment of the parents matches the future environment of their offspring.

Data accessibility. Data are available from the Dryad Digital Repository: https://doi.org/10.5061/dryad.fn2z34tt9 [56].

Authors' contributions. E.R.I.-C.: conceptualization, data curation, formal analysis, investigation, methodology, project administration, visualization, writing—original draft, and writing—review and editing; K.S.: conceptualization, data curation, investigation, methodology, writing—review and editing; H.C.: conceptualization, methodology, and writing—review and editing; S.I.: conceptualization, project administration, and writing—review and editing; T.C.: conceptualization, project administration, and writing—review and editing; A.A.M.: conceptualization, funding acquisition, project administration, supervision, writing—original draft, and writing—review and editing.

All authors gave final approval for publication and agreed to be held accountable for the work performed therein.

Competing interests. We declare we have no competing interests.

Funding. This work was funded by BBSRC BB/R017387/1 and ERC GermlineAgeingSoma 724909 to A.A.M.

Acknowledgements. Thanks go to members of the Maklakov Lab, Eryn McFarlane, Joel Pick and Josh Moatt for statistical advice.

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
