## [Peer Review File · Proceedings of the Royal Society B: Biological Sciences]

Review History

RSPB-2020-2889.R0 (Original submission)

Review form: Reviewer 1

Recommendation

Major revision is needed (please make suggestions in comments)

Scientific importance: Is the manuscript an original and important contribution to its field?

Good

General interest: Is the paper of sufficient general interest?

Good

Quality of the paper: Is the overall quality of the paper suitable?

Acceptable

Is the length of the paper justified?

Yes

Should the paper be seen by a specialist statistical reviewer?

No

Do you have any concerns about statistical analyses in this paper? If so, please specify them explicitly in your report.

Yes

It is a condition of publication that authors make their supporting data, code and materials available - either as supplementary material or hosted in an external repository. Please rate, if applicable, the supporting data on the following criteria.

Is it accessible?

Yes

Is it clear?

Yes

Is it adequate?

Yes

Do you have any ethical concerns with this paper?

No

Comments to the Author

Review of RSPB-2020-2889

This study addresses an important question in the evolutionary and functional biology of ageing: what are the transgenerational fitness / life-history consequences of dietary restriction (DR)? This is a major subject which the field does not yet understand very well, and the present study is a welcome addition to advancing our understanding of this issue. Specifically, the authors examine the effects of temporary fasting on mortality, reproduction and measures of fitness across 3 generations of descendants in the nematode *C. elegans*. Interestingly, the authors' results suggest that – amongst other things – great-grandparental exposure to temporary fasting reduces fitness and increases mortality risk of F3 descendants. These novel findings highlight the importance of transgenerational trade-offs involved in the life-history consequences of DR.

The subject is clearly important, and the authors present novel results that are worthwhile publishing. My major problem with the manuscript, as it is currently written, is the presentation of the methods and the results (in marked contrast, the abstract, introduction and discussion are clearly and well written).

First, the details of the experimental design are rather complicated and should be much more clearly explained so that the reader can follow more easily what was done and how, maybe using one or two figures (in the spirit of Fig. S1, which, however, has some problems; see below). In general, the methods section is very hard to follow. (Sometimes, it is even somewhat difficult to judge from the rather convoluted description whether there may be a problem with aspects of the experimental design or statistical analyses or not.)

Secondly, the results section is hard to follow and rather tedious to read, in part due to a lot of statistical details given in parentheses and in part due to the modular structure of P0...to F3, separated into survival and reproduction, etc. I was wondering whether there might not be a more effective way of conveying the results to the reader and telling a story that hangs together tightly, maybe by grouping the results by trait rather than by separately discussing each generation and trait and/or perhaps by relegating all statistical details given in parentheses to tables and figures. Another option would perhaps be to combine the Results and Discussion section and/or to use statements of main results as headers rather than P0, F1, etc. Overall, I felt that the results section would benefit greatly from being written in similar spirit as the discussion. The legends of the main figures could also be expanded so that they briefly summarize the main results presented in the subfigures/panels, and all abbreviations should be

defined here again so that the figures can be understood as stand-alone items. These things would help the reader greatly in terms of digesting and understanding the often times complex results.

In sum, I feel that this is a potentially interesting study reporting novel findings but the clarity of presentation could be much improved - this would make the paper much more readable and thus more impactful.

Minor/detailed comments:

Intro: the intro is generally clearly and well written; the questions are well laid out.

L39: temporary vs. intermittent fasting: difference?

L68: How does larval starvation differ from DR? This may not be clear to the generalist reader; in many animals starvation decreases lifespan, unlike DR, so some clarification may be necessary here.

Methods: in general, presentation/explanation could be improved; design rather complex and needs better explanation

L121, and elsewhere: 4 plates of 4 different types of plates? Not clear.

L121, and elsewhere: regarding the 4 treatments, how much replication was there for each of these treatments?

L121, and elsewhere. Similarly, how were the experimental blocks defined? Maybe indicate those, or at least mention them, also in Fig.S1?

I agree that a figure (or two, if necessary), in the spirit of Fig.S1, is the perhaps best way to convey the details of the experimental design. However, Fig.S1 does not do this job adequately (yet): there is lots of relevant information about the design missing from the legend of that figure. The legend of that figure should be expanded to include more detail of what is actually shown in the figure (e.g., the legends say "discussed below" - but I could not find any discussion in the supplementary file). Replication and blocks etc. should also be clarified here. The figure explaining the experimental design is quite central for understanding the experiments - maybe consider moving the figure into the main manuscript?

L141: Might this not lead to potential confounding effects? In some ageing studies (in rodents, etc.) it has been found that variation in "birth date" can have a significant impact on adult lifespan but is often unaccounted for in statistical models. Maybe a caveat should be included here?

L149-151. This explanation / rationale is insufficiently clear to me.

L161. Blocks. Maybe use a figure to clarify the design? Also see comments above.

L172-174. This is also a bit hard to understand - why was this done? What's the underlying rationale / reason?

L182. Incomplete sentence - reword.

L186-L190, and elsewhere. It might be best to give the actual statistical models as equations - this would be clearer and more explicit.

L186-L190. Issue replication mentioned further above; definition of blocks, etc. – as mentioned above, the experimental design as described in the M&M section is not sufficiently clear. A figure and an improved written description of the design would be important for the reader to grasp the details of the design.

L130. Similar or the same fixed and random effects? Vague. Perhaps better give the model equation?

Fig.1A-D legend. It should read “Cox” not “cox”.

L405-406. Why is there this difference? The reader is being told this fact but the generalist reader who does not know that previous paper might have a hard time understanding what this finding means vis-à-vis the findings reported here. Is this a discrepancy or not? If it is, how could it be explained? See above: how is larval starvation in *C. elegans* different (or not) from DR?

Review form: Reviewer 2

Recommendation

Accept with minor revision (please list in comments)

Scientific importance: Is the manuscript an original and important contribution to its field?

Good

General interest: Is the paper of sufficient general interest?

Acceptable

Quality of the paper: Is the overall quality of the paper suitable?

Good

Is the length of the paper justified?

Yes

Should the paper be seen by a specialist statistical reviewer?

No

Do you have any concerns about statistical analyses in this paper? If so, please specify them explicitly in your report.

Yes

It is a condition of publication that authors make their supporting data, code and materials available - either as supplementary material or hosted in an external repository. Please rate, if applicable, the supporting data on the following criteria.

Is it accessible?

Yes

Is it clear?

Yes

Is it adequate?

Yes

Do you have any ethical concerns with this paper?

No

Comments to the Author

Dietary restriction (DR) features prominently in the biology of aging. It has often been shown that organisms fed a restricted diet live longer than those fed an unrestricted diet, but that there are trade-offs with fecundity. One variety of dietary restriction involves temporary fasting (TF), which has clinical applications in humans. The effects of DR have been shown to carry over to the offspring and grand-offspring of individuals subjected to DR in a variety of organisms ("cross-generational" effects), and sometimes to the F3 and beyond ("transgenerational" effects). Various proximate (physiological) and ultimate (evolutionary) models have been posited to explain the phenomenon of DR-enhanced lifespan, with its (apparent) concomitant tradeoffs in fecundity.

The authors report a study using a study with the nematode model *C. elegans*, in which they investigate several questions concerning the cross-generational and transgenerational effects of TF on lifespan, fitness, and growth. Specifically, they address the questions (their words): "...addressing the following unresolved questions: 1) How does TF affect mortality risk and reproductive ageing once the animals return to their standard food regime?; 2) How do offspring of fasting parents perform in matching and mis-matching environments?; 3) Do transgenerational effects of ancestral fasting shape mortality risk and reproductive ageing of distant descendants?; and 4) Does reduced reproduction under DR represent a decision-making strategy?"

Worms are assigned to one of four dietary treatment groups: Ad libitum (AL), food odor (FO), AL+FO, and Temporary Fasting (TF). Individual lineages were carried out to the F3 generation (sometimes less, depending on the question of interest). Phenotypes of interest are lifespan, lifetime fecundity, individual fitness (the dominant eigenvalue of the Leslie matrix derived from the life-table), and body size measured at various life stages. Individual lineages can be assigned a three or four generation dietary state using the notation (P0-F1-F2) or (P0-F1-F2-F3), where the state at each generation is one of the four treatment groups (e.g., in the FO-TF-AL group, the parent was raised under the food odor treatment, the F1 in the temporary fasting treatment, and the F2 in the AL treatment). From this design, the generation-specific effects of treatment can be quantified, and the fitness effects assessed.

This a very elegant experiment, and the results are admirably transparent. I think the interpretation is largely sensible. I do have a quibble with the authors' terminology, which is that they use the terms "positive" and "beneficial" seemingly interchangeably, and likewise the terms "negative" and "deleterious". "Positive" and "negative" are straightforwardly interpreted as an increase or decrease in trait value, even when the trait value is the dominant eigenvalue of the Leslie matrix ("individual fitness", which is a sensible definition). However, "beneficial" implies "favored by natural selection" (and similarly "deleterious"=disfavored), and in the absence of theoretical analysis (or a selection experiment), it's impossible to say what would be favored by natural selection. However, I think that the authors' have shown pretty clearly that cross-generational effects that are unambiguously positive may be negative in further generations in particular circumstances.

I have one non-trivial criticism, which concerns the presentation of the analysis. The model(s) needs to be written out explicitly, preferably in math or at the very least in pseudo-code. The interested reader should be able to re-create the analysis, and unless one is familiar with the specific R packages in question, that does not seem possible (and probably not even then, I bet).

Also, Figure S1 needs to be Figure 1 in the main text. I could not make it through the results without constantly referring to the figure.

Some minor comments, by Page/line.

P8/L149. Should be Figure S1 (?)

P12/L247 (and throughout). The authors have a tendency to use syntax that is the opposite of what is shown in a figure. Here, they say "...offspring show decreased mortality when placed in AL conditions". Which is true, but the figure shows the AL treatments on the positive side. If

you say "increased survivorship", the human brain (or at least my brain) has an easier time of it.

P15/L300. I like the (P0-F1-F2) notation, but it could be interpreted the other way around, and the reader has to do more thinking than necessary. Write out the interpretation (i.e., just like I wrote it) the first time you use it.

P15/L314. "...TF grandparents with AL parents (FO-TF-TF:...)" I think should be ...(FO-AL-TF).

P15/L316. from the sentence, should this be reversed, i.e., AL-AL-TF? But you reported AL-AL-TF above (line 312). Something is not right here.

Decision letter (RSPB-2020-2889.R0)

23-Dec-2020

Dear Dr Ivimey-Cook:

I am writing to inform you that your manuscript RSPB-2020-2889 entitled "Transgenerational fitness effects of lifespan extension by dietary restriction in *Caenorhabditis elegans*" has, in its current form, been rejected for publication in Proceedings B.

This action has been taken on the advice of referees, who have recommended that substantial revisions are necessary. With this in mind we would be happy to consider a resubmission, provided the comments of the referees are fully addressed. However please note that this is not a provisional acceptance.

Sincerely,
 Dr Maurine Neiman
 mailto: proceedingsb@royalsociety.org

Associate Editor

Board Member: 1

Comments to Author:

This is an interesting paper looking at the effects of temporary fasting on fitness components in the same and up to three downstream generations in *C. elegans*. Transgenerational effects of dietary restriction have been reported in this system before, but this paper is a substantial contribution in that it (1) uses a specific form of dietary restriction, namely temporary fasting in the juvenile stage, which has been recently found to have similar effects as "classic" dietary restriction, also in rodents, (2) attempts to disentangle the effects of diet in the current and previous generations, as well as the effects of sensory signals, in an (incomplete) factorial design, and (3) presents results on four consecutive generations. The most interesting finding is that temporary fasting has a negative effect on grandoffspring and great-grandoffspring. Both reviewers found this study interesting; trade-offs between within-generation and transgenerational effects dietary restriction throw a new perspective on the effect of diet on fitness and lifespan. Both, however, raise issues with the description of the design and analysis, and the reporting of results.

I agree with this assessment. Indeed the design is hard to follow and results very tedious. This is in part because of the complexity of the study, but also in part because the specific questions that those results could address are not clearly formulated. In addition, I have some major issues with the analysis, which are in part linked with the issue of the design and which questions are being addressed.

- While four specific questions are formulated at the end of the Introduction, neither the Results nor the Discussion are structured along those questions. Rather, Results describe trait by trait and generation by generation the different pairwise comparisons of treatments, in a spirit of "let's see what the effects of the plethora of treatments might be in generation P0, F1, etc.". This way of thinking is apparent in the analysis – even though the design is factorial (effect of diet vs odor for P0, parent x offspring diet for F1, grand parent x parent x offspring diet for F2 etc), the analysis flattens this design into a one-way 8-level classification. Furthermore, it is limited to reporting the parameters for the model that tell us how each of the seven treatments deviate from the "baseline" ad lib treatment, and some post-hoc pairwise comparisons apparently based on a difference between the parameter values (e.g., l. 238, 240). The associated P values do not seem to have been corrected for multiple comparisons. But more importantly, they do not seem to be guided by the questions or predictions, making the results very tedious to follow and ultimately less interesting that they could be.

- The results are reported in terms of parameter values and in some cases differences in parameter values. It is very hard to make biological sense of them. E.g., the authors state (l. 237-8) that the FO worms produced a "far greater" number of offspring than TF, and they support this statement with a value of $\beta = 1.99$. What is a definition of "far greater"? What are the units of β ? It is obviously not 1.99 offspring more on average, but how do I translate the 1.99 in something that I can biologically interpret? What the authors should report is estimated marginal means; I would like to see statement like "FO worms produced on average X % offspring more (Y vs Z offspring) than TF worms."

- Question 2 is explicitly about the interaction between the parental and offspring treatment, but his interaction is not explicitly tested. Still, the authors imply that the effect of parental diet are contingent on offspring diet (l.244ff). In fact, looking at figure 1B, the effects of the parental and offspring treatment on offspring survival seem perfectly additive. For reproduction the interaction is indirectly supported by the effect of parental dietary treatment being of opposite sign depending on offspring treatment, but I would still like interaction to be properly supported with a likelihood ratio test.

- I see an interest to include the food odor treatment, but it is not related to any of the four specific questions (or if it is, I missed it). The existence of FO... and AL+FO... treatments actually adds confusion to understanding of the results of the "main" treatments. So it would make sense to separate them in terms of reporting, analysis and discussion – as long as the authors specify a question to which they are relevant.

- From generation F2 the design ceases to be fully factorial – not all F1 or F2 treatments are applied in combination with all P0 treatments. I understand that trying to do a full factorial over four generations would explode the design, but no rationale is given for choosing the specific treatment combinations. For example, why is treatment TF-TF-AL (or TF-TF-TF-AL) not included?

- The authors use Cox regression to compare survival and only report the parameters estimated in that model. The actual data in the form of survival curves should be presented. Furthermore, the authors should verify the appropriateness of Cox regression (i.e., proportional hazard). What I am wondering, based on the shift in egg laying pattern, is whether the lifespan difference is not mainly explained by extended juvenile/pre-reproductive phase. In fact, the authors mention this (l. 441), but never address head on. What is the effect of the treatments on developmental time. Would the survival curves actually look the same if reproductive maturation were taken as the starting point? Also, it would be interesting to see if the demographic nature of the change in mortality is the same for within- and transgenerational effects. One could in principle imagine that TF in P0 results in slowed aging in P0 but increase juvenile mortality in F2 and F3.

- The figures are highly confusing. The acronyms with all their plus and minus signs are hard to decipher, in particular given the small font and poor resolution. The same colors are used to indicate different treatments in graphs that otherwise look the same. E.g. the treatment in pink in fig 1 and 2 is fed ad libitum in generation P0, but the same color in fig 3 and 4 indicates a treatment which is only exposed to food odor in P0. Conversely, the results from the same treatment are in part indicated with different colors. In Fig 1A vs 1C it is just the shade of the color that is different. But, e.g., in Fig 3 FO-TF-TF is pink but in Fig 4 it is brown, and pink is used for a different treatment - even though the two plots report data from the same generation in the same experiment. The order of treatment in fig 3 and 4 is different. The colors for FO+AL-AL-AL and AL-AL-TF in Fig 3 are indistinguishable. The treatments are alternately called, e.g., "AL-AL-TF" and "AL-AL in TF". It almost seems like the authors made an effort to make the reader's life more difficult.

- I agree with the reviewers that Fig S1 should be a main figure. I am also wondering if the "icons" for the treatments could be made more self-explanatory. I first thought that the bottom orange disk in FO+AL treatment represents food and the top orange disk the odor that was somehow applied from above. Which made understanding of the other treatments difficult. Finally, it does not help that the order of treatments in the figures does not correspond to the order in which they are described in the text.

Other comments:

- The technical part of methods are also hard to follow for a non-worm reader. What is substitute food source? What is the role of peptone? What is meant by bleaching? I understand these may be obvious for people working with *C. elegans*, but the authors should aim for a general audience.

The "Concluding Remarks" do not seem to add anything that has not been said in the Discussion just - above, they should be removed.

- l. 206; lambda describes a multiplicative process, so it would make sense to use its log for the analysis.

- l. 208: I thought the TF treatment increased development time – why was the same interval added for all treatments?

- l. 238, 240 and other places: the reader is sent to consult multiple supplementary tables, which actually do not contain the value reported in the text. It took me some time to figure out that the "beta" values reported are differences between parameter values, I still don't know how the SE (or whatever the number in parentheses is) was calculated, given that the parameter estimates are not independent. E.g., the value reported in l. 238 should be referred to as FO-TF (and not as FO).

- l. 414: This is formulated as a statement about results ("we found") but no data measuring somatic maintenance are presented in the paper.

To summarize, this paper needs a major restructuring, some reanalysis, a change of the reporting of results to one that comes closer to a biological interpretation, both in terms of size of effects and underlying questions.

Reviewer(s)' Comments to Author:

Referee: 1

Comments to the Author(s)

Review of RSPB-2020-2889

This study addresses an important question in the evolutionary and functional biology of ageing: what are the transgenerational fitness / life-history consequences of dietary restriction (DR)? This is a major subject which the field does not yet understand very well, and the present study is a welcome addition to advancing our understanding of this issue. Specifically, the authors examine the effects of temporary fasting on mortality, reproduction and measures of fitness across 3 generations of descendants in the nematode *C. elegans*. Interestingly, the authors' results suggest that – amongst other things – great-grandparental exposure to temporary fasting reduces fitness and increases mortality risk of F3 descendants. These novel findings highlight the importance of transgenerational trade-offs involved in the life-history consequences of DR.

The subject is clearly important, and the authors present novel results that are worthwhile publishing. My major problem with the manuscript, as it is currently written, is the presentation of the methods and the results (in marked contrast, the abstract, introduction and discussion are clearly and well written).

First, the details of the experimental design are rather complicated and should be much more clearly explained so that the reader can follow more easily what was done and how, maybe using one or two figures (in the spirit of Fig. S1, which, however, has some problems; see below). In general, the methods section is very hard to follow. (Sometimes, it is even somewhat difficult to judge from the rather convoluted description whether there may be a problem with aspects of the experimental design or statistical analyses or not.)

Secondly, the results section is hard to follow and rather tedious to read, in part due to a lot of statistical details given in parentheses and in part due to the modular structure of P0... to F3, separated into survival and reproduction, etc. I was wondering whether there might not be a more effective way of conveying the results to the reader and telling a story that hangs together tightly, maybe by grouping the results by trait rather than by separately discussing each generation and trait and/or perhaps by relegating all statistical details given in parentheses to tables and figures. Another option would perhaps be to combine the Results and Discussion section and/or to use statements of main results as headers rather than P0, F1, etc. Overall, I felt that the results section would benefit greatly from being written in similar spirit as the discussion. The legends of the main figures could also be expanded so that they briefly summarize the main results presented in the subfigures/panels, and all abbreviations should be defined here again so that the figures can be understood as stand-alone items. These things

would help the reader greatly in terms of digesting and understanding the often times complex results.

In sum, I feel that this is a potentially interesting study reporting novel findings but the clarity of presentation could be much improved - this would make the paper much more readable and thus more impactful.

Minor/detailed comments:

Intro: the intro is generally clearly and well written; the questions are well laid out.

L39: temporary vs. intermittent fasting: difference?

L68: How does larval starvation differ from DR? This may not be clear to the generalist reader; in many animals starvation decreases lifespan, unlike DR, so some clarification may be necessary here.

Methods: in general, presentation/explanation could be improved; design rather complex and needs better explanation

L121, and elsewhere: 4 plates of 4 different types of plates? Not clear.

L121, and elsewhere: regarding the 4 treatments, how much replication was there for each of these treatments?

L121, and elsewhere. Similarly, how were the experimental blocks defined? Maybe indicate those, or at least mention them, also in Fig.S1?

I agree that a figure (or two, if necessary), in the spirit of Fig.S1, is the perhaps best way to convey the details of the experimental design. However, Fig.S1 does not do this job adequately (yet): there is lots of relevant information about the design missing from the legend of that figure. The legend of that figure should be expanded to include more detail of what is actually shown in the figure (e.g., the legends say "discussed below" - but I could not find any discussion in the supplementary file). Replication and blocks etc. should also be clarified here. The figure explaining the experimental design is quite central for understanding the experiments - maybe consider moving the figure into the main manuscript?

L141: Might this not lead to potential confounding effects? In some ageing studies (in rodents, etc.) it has been found that variation in "birth date" can have a significant impact on adult lifespan but is often unaccounted for in statistical models. Maybe a caveat should be included here?

L149-151. This explanation / rationale is insufficiently clear to me.

L161. Blocks. Maybe use a figure to clarify the design? Also see comments above.

L172-174. This is also a bit hard to understand - why was this done? What's the underlying rationale / reason?

L182. Incomplete sentence - reword.

L186-L190, and elsewhere. It might be best to give the actual statistical models as equations - this would be clearer and more explicit.

L186-L190. Issue replication mentioned further above; definition of blocks, etc. - as mentioned above, the experimental design as described in the M&M section is not sufficiently clear. A figure

and an improved written description of the design would be important for the reader to grasp the details of the design.

L130. Similar or the same fixed and random effects? Vague. Perhaps better give the model equation?

Fig.1A-D legend. It should read "Cox" not "cox".

L405-406. Why is there this difference? The reader is being told this fact but the generalist reader who does not know that previous paper might have a hard time understanding what this finding means vis-à-vis the findings reported here. Is this a discrepancy or not? If it is, how could it be explained? See above: how is larval starvation in *C. elegans* different (or not) from DR?

Referee: 2

Comments to the Author(s)

Dietary restriction (DR) features prominently in the biology of aging. It has often been shown that organisms fed a restricted diet live longer than those fed an unrestricted diet, but that there are trade-offs with fecundity. One variety of dietary restriction involves temporary fasting (TF), which has clinical applications in humans. The effects of DR have been shown to carry over to the offspring and grand-offspring of individuals subjected to DR in a variety of organisms ("cross-generational" effects), and sometimes to the F3 and beyond ("transgenerational" effects). Various proximate (physiological) and ultimate (evolutionary) models have been posited to explain the phenomenon of DR-enhanced lifespan, with its (apparent) concomitant tradeoffs in fecundity.

The authors report a study using a study with the nematode model *C. elegans*, in which they investigate several questions concerning the cross-generational and transgenerational effects of TF on lifespan, fitness, and growth. Specifically, they address the questions (their words): "...addressing the following unresolved questions: 1) How does TF affect mortality risk and reproductive ageing once the animals return to their standard food regime?; 2) How do offspring of fasting parents perform in matching and mis-matching environments?; 3) Do transgenerational effects of ancestral fasting shape mortality risk and reproductive ageing of distant descendants?; and 4) Does reduced reproduction under DR represent a decision-making strategy?"

Worms are assigned to one of four dietary treatment groups: Ad libitum (AL), food odor (FO), AL+FO, and Temporary Fasting (TF). Individual lineages were carried out to the F3 generation (sometimes less, depending on the question of interest). Phenotypes of interest are lifespan, lifetime fecundity, individual fitness (the dominant eigenvalue of the Leslie matrix derived from the life-table), and body size measured at various life stages. Individual lineages can be assigned a three or four generation dietary state using the notation (P0-F1-F2) or (P0-F1-F2-F3), where the state at each generation is one of the four treatment groups (e.g., in the FO-TF-AL group, the parent was raised under the food odor treatment, the F1 in the temporary fasting treatment, and the F2 in the AL treatment). From this design, the generation-specific effects of treatment can be quantified, and the fitness effects assessed.

This a very elegant experiment, and the results are admirably transparent. I think the interpretation is largely sensible. I do have a quibble with the authors' terminology, which is that they use the terms "positive" and "beneficial" seemingly interchangeably, and likewise the terms "negative" and "deleterious". "Positive" and "negative" are straightforwardly interpreted as an increase or decrease in trait value, even when the trait value is the dominant eigenvalue of the Leslie matrix ("individual fitness", which is a sensible definition). However, "beneficial" implies "favored by natural selection" (and similarly "deleterious"=disfavored), and in the absence of theoretical analysis (or a selection experiment), it's impossible to say what would be favored by natural selection. However, I think that the authors' have shown pretty clearly that cross-generational effects that are unambiguously positive may be negative in further generations in particular circumstances.

I have one non-trivial criticism, which concerns the presentation of the analysis. The model(s) needs to be written out explicitly, preferably in math or at the very least in pseudo-code. The interested reader should be able to re-create the analysis, and unless one is familiar with the specific R packages in question, that does not seem possible (and probably not even then, I bet). Also, Figure S1 needs to be Figure 1 in the main text. I could not make it through the results without constantly referring to the figure.

Some minor comments, by Page/line.

P8/L149. Should be Figure S1 (?)

P12/L247 (and throughout). The authors have a tendency to use syntax that is the opposite of what is shown in a figure. Here, they say "...offspring show decreased mortality when placed in AL conditions". Which is true, but the figure shows the AL treatments on the positive side. If you say "increased survivorship", the human brain (or at least my brain) has an easier time of it.

P15/L300. I like the (P0-F1-F2) notation, but it could be interpreted the other way around, and the reader has to do more thinking than necessary. Write out the interpretation (i.e., just like I wrote it) the first time you use it.

P15/L314. "...TF grandparents with AL parents (FO-TF-TF:...)" I think should be ...(FO-AL-TF).

P15/L316. from the sentence, should this be reversed, i.e., AL-AL-TF? But you reported AL-AL-TF above (line 312). Something is not right here.

Author's Response to Decision Letter for (RSPB-2020-2889.R0)

See Appendix A.

RSPB-2021-0701.R0

Review form: Reviewer 2

Recommendation

Accept with minor revision (please list in comments)

Scientific importance: Is the manuscript an original and important contribution to its field?

Good

General interest: Is the paper of sufficient general interest?

Good

Quality of the paper: Is the overall quality of the paper suitable?

Good

Is the length of the paper justified?

Yes

Should the paper be seen by a specialist statistical reviewer?

No

Do you have any concerns about statistical analyses in this paper? If so, please specify them explicitly in your report.

No

It is a condition of publication that authors make their supporting data, code and materials available - either as supplementary material or hosted in an external repository. Please rate, if applicable, the supporting data on the following criteria.

Is it accessible?

Yes

Is it clear?

Yes

Is it adequate?

Yes

Do you have any ethical concerns with this paper?

No

Comments to the Author

The authors have satisfactorily addressed the comments I provided in my previous review. The other reviewer noted that the authors should present a figure with the survivorship curves, which was a good idea. Those figures (S4) are not referenced in the paper. In fact, I think the survivorship curves should replace the line diagrams of the odds ratios in Fig. 2, which don't add anything beyond the values reported in the text.

Decision letter (RSPB-2021-0701.R0)

12-Apr-2021

Dear Dr Ivimey-Cook:

Your manuscript has now been peer reviewed and the reviews have been assessed by an Associate Editor. The reviewers' comments (not including confidential comments to the Editor) and the comments from the Associate Editor are included at the end of this email for your reference. As you will see, the reviewers and the Editors have raised some concerns with your manuscript and we would like to invite you to revise your manuscript to address them.

When submitting your revision please upload a file under "Response to Referees" in the "File Upload" section. This should document, point by point, how you have responded to the

reviewers' and Editors' comments, and the adjustments you have made to the manuscript. We require a copy of the manuscript with revisions made since the previous version marked as 'tracked changes' to be included in the 'response to referees' document.

Research ethics:

Use of animals and field studies:

It is a condition of publication that you make available the data and research materials supporting the results in the article (<https://royalsociety.org/journals/authors/author-guidelines/#data>). Datasets should be deposited in an appropriate publicly available repository and details of the associated accession number, link or DOI to the datasets must be included in the Data Accessibility section of the article (<https://royalsociety.org/journals/ethics-policies/data-sharing-mining/>). Reference(s) to datasets should also be included in the reference list of the article with DOIs (where available).

Online supplementary material will also carry the title and description provided during submission, so please ensure these are accurate and informative. Note that the Royal Society will not edit or typeset supplementary material and it will be hosted as provided. Please ensure that

the supplementary material includes the paper details (authors, title, journal name, article DOI). Your article DOI will be 10.1098/rspb.[paper ID in form xxxx.xxxx e.g. 10.1098/rspb.2016.0049].

Please submit a copy of your revised paper within three weeks. If we do not hear from you within this time your manuscript will be rejected. If you are unable to meet this deadline please let us know as soon as possible, as we may be able to grant a short extension.

Best wishes,
Dr Maurine Neiman
mailto:proceedingsb@royalsociety.org

Associate Editor
Comments to Author:

I obtained one review for this resubmitted version as well as having read it carefully myself. I agree with the reviewer that the revision reads better and the results are interpreted in a more interesting way in the Discussion. I also agree with reviewer 1 that the odds plots should be replaced by survival curves; the odds ratios are anyway given in the text.

I have one fairly substantial issue that still needs to be addresses. I did not raise it in the previous round of reviews because I only noticed the relevant piece of information at this reading. I apologize for what may seem like moving goalposts, but I feel partly excused because this information is rather buried in the Methods. Namely, the authors mentions "disproportionate day one and two mortality of TF individuals" (l. 154). They bring this up in the context of the sample sizes, but I think this – apparently unusually high – mortality under the temporary fasting treatment has important implications for interpretation of the results.

First, it seem to contradict what is said in l. 439ff about the TF treatment not resulting in malnutrition. If the TF did not cause malnutrition, why were these young worms dying?

Second, this opens the possibility that this mortality might have been non-random and thus may have imposed some selection, the consequences of which might have contributed to the results. Could some of the trans-generational effects be due to genetically- based response to selection? How genetically variable was the strain used? Even if the strain was isogenic (info on genetic variation of the strain is not provided), could it be that TF preferentially eliminated phenotypically frail individuals that would otherwise be first to die of old age, thus biasing the lifespan upwards?

The paper needs be more upfront with this issue. The information about the magnitude of this mortality during the TF and other treatments should be provided and the issue of potential selection confound brought up in the Discussion. I the authors have arguments as to why this mortality is unlikely to have biased the results, these arguments should be laid out in Discussion; otherwise, the caveats should be made explicit.

Except for this point, I have a few minor editorial suggestions:

l. 28: "in the parental generation": to minimize risk of confusion, I would rephrase "of the individuals subject to TF" ("parental" refers to parents to some focal individuals, which are not clear here), and then I would replace "future generations" by "their descendants" (as only descendants of the treated individuals are affected and not the entire future generations). Surely, most evolutionary biologists would have no problem with understanding the current version correctly, but abstracts can be read by a wide audience, so it is good to be very precise.

l. 88-89: it is not clear from the sentence structure if the clause "suggesting that..." refers to the effect of DR or the effect of food odor reversing the effects of DR

l. 96 "distant" => "more distant" or "descendants beyond offspring"?

l. 97: I am not convinced that the design would have allowed to discern whether or not this is a "decision-making strategy", in particular in the absence of any operational definition of "strategy". It would be better to rephrase this question in terms closer to what is actually shown, i.e. , response mediated by odor perception (or absence thereof).

l. 470: "integral to this increase" : meaning what exactly?

Reviewer(s)' Comments to Author:

Referee: 2

Comments to the Author(s).

The authors have satisfactorily addressed the comments I provided in my previous review. The other reviewer noted that the authors should present a figure with the survivorship curves, which was a good idea. Those figures (S4) are not referenced in the paper. In fact, I think the survivorship curves should replace the line diagrams of the odds ratios in Fig. 2, which don't add anything beyond the values reported in the text.

Author's Response to Decision Letter for (RSPB-2021-0701.R0)

See Appendix B.

Decision letter (RSPB-2021-0701.R1)

15-Apr-2021

Dear Dr Ivimey-Cook

I am pleased to inform you that your manuscript entitled "Transgenerational fitness effects of lifespan extension by dietary restriction in *Caenorhabditis elegans*" has been accepted for publication in *Proceedings B*.

Data Accessibility section

Open Access

Paper charges

Sincerely,

Dr Maurine Neiman

Associate Editor:

Board Member

Comments to Author:

I am pleased with the revisions and happy to recommend acceptance of this paper.

Tadeusz Kawecki

Appendix A

Associate Editor

Board Member: 1

Comments to Author:

This is an interesting paper looking at the effects of temporary fasting on fitness components in the same and up to three downstream generations in *C. elegans*. Transgenerational effects of dietary restriction have been reported in this system before, but this paper is a substantial contribution in that it (1) uses a specific form of dietary restriction, namely temporary fasting in the juvenile stage, which has been recently found to have similar effects as "classic" dietary restriction, also in rodents, (2) attempts to disentangle the effects of diet in the current and previous generations, as well as the effects of sensory signals, in an (incomplete) factorial design, and (3) presents results on four consecutive generations. The most interesting finding is that temporary fasting has a negative effect on grandoffspring and great-grandoffspring. Both reviewers found this study interesting; trade-offs between within-generation and transgenerational effects dietary restriction throw a new perspective on the effect of diet on fitness and lifespan. Both, however, raise issues with the description of the design and analysis, and the reporting of results.

I agree with this assessment. Indeed the design is hard to follow and results very tedious. This is in part because of the complexity of the study, but also in part because the specific questions that those results could address are not clearly formulated. In addition, I have some major issues with the analysis, which are in part linked with the issue of the design and which questions are being addressed.

We appreciate this comment and have addressed each one of the valuable comments below. As such we believe the manuscript has improved.

-While four specific questions are formulated at the end of the Introduction, neither the Results nor the Discussion are structured along those questions. Rather, Results describe trait by trait and generation by generation the different pairwise comparisons of treatments, in a spirit of "let's see what the effects of the plethora of treatments might be in generation P0, F1, etc.". This way of thinking is apparent in the analysis – even though the design is factorial (effect of diet vs odor for P0, parent x offspring diet for F1, grand parent x parent x offspring diet for F2 etc), the analysis flattens this design into a one-way 8-level classification. Furthermore, it is limited to reporting the parameters for the model that tell us how each of the seven treatments deviate from the "baseline" ad lib treatment, and some post-hoc pairwise comparisons apparently based on a difference between the parameter values (e.g., l. 238, 240). The associated P values do not seem to have been corrected for multiple comparisons. But more importantly, they do not seem to be guided by the questions or predictions, making the results very tedious to follow and ultimately less interesting that they could be.

We agree with this comment, and as such have reframed the analysis in the F1 generation to specifically answer Question 2 by looking at the interaction between parent/offspring diet. As the other questions can be answered specifically by looking the effects of treatment (Question 1), Great-grandparental treatment (Question 3) and FO (Food Odour) effects (Question 4) we believe we have presented the results and discussion in such a way that these questions are answered, and all the results are transparent. While we agree the results aren't laid out as suggested, trait by trait, we wanted to ensure enough transparency of diet effects generation by generation. In particular, focusing on the trade-offs between survival and reproduction at each generation. As comparisons involved were based on planned contrasts we didn't believe that accounting for multiple comparisons was appropriate, specifically as the vast majority of pairwise comparisons were not relevant to the questions. The main comparisons were involving baseline AL (and subsequent ancestral generations of successive AL) or the equivalent in TF.

- The results are reported in terms of parameter values and in some cases differences in parameter values. It is very hard to make biological sense of them. E.g., the authors state (l. 237-8) that the FO worms produced a "far greater" number of offspring than TF, and they support this statement with a value of $\beta = 1.99$. What is a definition of "far greater"? What are the units of β ? It is obviously not 1.99 offspring more on average, but how do I translate the 1.99 in something that I can biologically interpret? What the authors should report is estimated marginal means; I would like to see statement like "FO worms produced on average X % offspring more (Y vs Z offspring) than TF worms."

This is an important point and we have since edited each estimate to reflect this valuable comment by reporting estimate marginal means from the “emmeans” package. Each statistic has been back-transformed and a mean value with a ratio is presented. We hope this increases the understanding of the relevant differences.

- Question 2 is explicitly about the interaction between the parental and offspring treatment, but this interaction is not explicitly tested. Still, the authors imply that the effect of parental diet are contingent on offspring diet (l.244ff). In fact, looking at figure 1B, the effects of the parental and offspring treatment on offspring survival seem perfectly additive. For reproduction the interaction is indirectly supported by the effect of parental dietary treatment being of opposite sign depending on offspring treatment, but I would still like interaction to be properly supported with a likelihood ratio test. **We agree and have since reframed the analysis to focus specifically on this interaction (see Lines 219-222).**

- I see an interest to include the food odor treatment, but it is not related to any of the four specific questions (or if it is, I missed it). The existence of FO... and AL+FO... treatments actually adds confusion to understanding of the results of the "main" treatments. So it would make sense to separate them in terms of reporting, analysis and discussion – as long as the authors specify a question to which they are relevant.

This is a valid point and we agree that we could have made the need for all four treatments clearer. We have expanded on the fourth question (Lines 101-102) to incorporate the absence of pheromones which precludes the comparison between food odour (DR with pheromones) and TF (DR without pheromones).

- From generation F2 the design ceases to be fully factorial – not all F1 or F2 treatments are applied in combination with all P0 treatments. I understand that trying to do a full factorial over four generations would explode the design, but no rationale is given for choosing the specific treatment combinations. For example, why is treatment TF-TF-AL (or TF-TF-TF-AL) not included?

We appreciate the comment that a fully-factorial experiment would have exploded the design. As mentioned on Lines 173-176, we focused on certain lineages in order to test the specific questions listed, increase sample size of the tested combinations and also due to the lack of a *priori* expectations. In this particular case, TF-TF-AL and TF-TF-TF-AL were not given consideration due to lack of a *priori* reasoning.

- The authors use Cox regression to compare survival and only report the parameters estimated in that model. The actual data in the form of survival curves should be presented. Furthermore, the authors should verify the appropriateness of Cox regression (i.e., proportional hazard).

This is an important point. In all cases, we have changed the Cox regression to an “Event History” analysis as the F3 survival data didn’t meet the assumptions of proportional hazards, which we outline in the methods (Lines 228-238). These two methodologies are qualitatively similar, and the results remain unchanged. We have added the survival curves in the supplementary (Fig S4A-D).

What I am wondering, based on the shift in egg laying pattern, is whether the lifespan difference is not mainly explained by extended juvenile/pre-reproductive phase. In fact, the authors mention this (l. 441), but never address head on. What is the effect of the treatments on developmental time. Would the survival curves actually look the same if reproductive maturation were taken as the starting point? Also, it would be interesting to see if the demographic nature of the change in mortality is the same for within- and transgenerational effects. One could in principle imagine that TF in P0 results in slowed aging in P0 but increase juvenile mortality in F2 and F3.

We agree that this would be interesting, but reproductive maturation is delayed in DR individuals by several hours (which is shown in Figure S3A). Fecundity and lifespan are measured in days, so we wouldn’t be able to see this difference in a demographic or age-specific sense.

- The figures are highly confusing. The acronyms with all their plus and minus signs are hard to decipher, in particular given the small font and poor resolution. The same colors are used to indicate different treatments in graphs that otherwise look the same. E.g. the treatment in pink in fig 1 and 2 is fed ad libitum in generation P0, but the same color in fig 3 and 4 indicates a treatment which is only exposed to food odor in P0. Conversely, the results from the same treatment are in part indicated with

different colors. In Fig 1A vs 1C it is just the shade of the color that is different. But, e.g., in Fig 3 FO-TF-TF is pink but in Fig 4 it is brown, and pink is used for a different treatment - even though the two plots report data from the same generation in the same experiment. The order of treatment in fig 3 and 4 is different. The colors for FO+AL-AL-AL and AL-AL-TF in Fig 3 are indistinguishable. The treatments are alternately called, e.g., "AL-AL-TF" and "AL-AL in TF". It almost seems like the authors made an effort to make the reader's life more difficult.

We agree that this could have been improved and as such, we have changed the colours of the plots to be more distinct and hopefully increased the resolution and ease of reading for all of the graphs. Furthermore, to reduce the number of hyphens and pluses we have changed FO+AL to AL(FO) and FO to TF(FO) to increase ease of reading.

- I agree with the reviewers that Fig S1 should be a main figure. I am also wondering if the "icons" for the treatments could be made more self-explanatory. I first thought that the bottom orange disk in FO+AL treatment represents food and the top orange disk the odor that was somehow applied from above. Which made understanding of the other treatments difficult. Finally, it does not help that the order of treatments in the figures does not correspond to the order in which they are described in the text.

This has since been added with far more detail- incorporating sample size for each level and specific information regarding dietary paradigm.

Other comments:

- The technical part of methods are also hard to follow for a non-worm reader.

What is substitute food source?

What is the role of peptone?

What is meant by bleaching?

I understand these may be obvious for people working with *C. elegans*, but the authors should aim for a general audience.

We have added more detail to these parts

The "Concluding Remarks" do not seem to add anything that has not been said in the Discussion just - above, they should be removed.

Removed.

- l. 206; lambda describes a multiplicative process, so it would make sense to use its log for the analysis.

We disagree with this suggestion. As lambda is typically normally distributed anyway, log-transforming this variable will have little impact particularly as there are often an abundance of individuals with zero lambda values (which when log-transformed would produce infinity). Furthermore, residuals of the model were checked and seemed to fit a normal distribution better when the response had not been transformed.

- l. 208: I thought the TF treatment increased development time – why was the same interval added for all treatments?

This is a good point. But, as the difference in development time is in hours and the age-specified matrix has time intervals in days, this can't be translated into a meaningful difference in the Leslie matrix.

- l. 238, 240 and other places: the reader is sent to consult multiple supplementary tables, which actually do not contain the value reported in the text. It took me some time to figure out that the "beta" values reported are differences between parameter values, I still don't know how the SE (or whatever the number in parentheses is) was calculated, given that the parameter estimates are not independent. E.g., the value reported in l. 238 should be referred to as FO-TF (and not as FO).

We have updated the supplementary material to incorporate all of the values reported in the text. Furthermore, we have replaced the beta values with more succinct information regarding test statistics.

- l. 414: This is formulated as a statement about results ("we found") but no data measuring somatic maintenance are presented in the paper.

Removed.

To summarize, this paper needs a major restructuring, some reanalysis, a change of the reporting of results to one that comes closer to a biological interpretation, both in terms of size of effects and underlying questions.

We appreciate the thoughtful comments and hope to have addressed all of the important suggestions given by the associate editor.

Reviewer(s)' Comments to Author:

Referee: 1

Comments to the Author(s)
Review of RSPB-2020-2889

This study addresses an important question in the evolutionary and functional biology of ageing: what are the transgenerational fitness / life-history consequences of dietary restriction (DR)? This is a major subject which the field does not yet understand very well, and the present study is a welcome addition to advancing our understanding of this issue. Specifically, the authors examine the effects of temporary fasting on mortality, reproduction and measures of fitness across 3 generations of descendants in the nematode *C. elegans*. Interestingly, the authors' results suggest that – amongst other things – great-grandparental exposure to temporary fasting reduces fitness and increases mortality risk of F3 descendants. These novel findings highlight the importance of transgenerational trade-offs involved in the life-history consequences of DR.

The subject is clearly important, and the authors present novel results that are worthwhile publishing. My major problem with the manuscript, as it is currently written, is the presentation of the methods and the results (in marked contrast, the abstract, introduction and discussion are clearly and well written).

Thanks! And we hope to have addressed the various concerns raised!

First, the details of the experimental design are rather complicated and should be much more clearly explained so that the reader can follow more easily what was done and how, maybe using one or two figures (in the spirit of Fig. S1, which, however, has some problems; see below). In general, the methods section is very hard to follow. (Sometimes, it is even somewhat difficult to judge from the rather convoluted description whether there may be a problem with aspects of the experimental design or statistical analyses or not.)

We appreciate the various comments relating to improving the methods. As a result, we believe this section has improved

Secondly, the results section is hard to follow and rather tedious to read, in part due to a lot of statistical details given in parentheses and in part due to the modular structure of P0...to F3, separated into survival and reproduction, etc. I was wondering whether there might not be a more effective way of conveying the results to the reader and telling a story that hangs together tightly, maybe by grouping the results by trait rather than by separately discussing each generation and trait and/or perhaps by relegating all statistical details given in parentheses to tables and figures. Another option would perhaps be to combine the Results and Discussion section and/or to use statements of main results as headers rather than P0, F1, etc. Overall, I felt that the results section would benefit greatly from being written in similar spirit as the discussion. The legends of the main figures could also be expanded so that they briefly summarize the main results presented in the subfigures/panels, and all abbreviations should be defined here again so that the figures can be understood as stand-alone items. These things would help the reader greatly in terms of digesting and understanding the often times complex results.

We appreciate the thought that went into this comment. However, we believe that in order to preserve the transparency and detail of the many results that we found, this format, however tedious, is the most appropriate. Grouping by trait was initially considered but as our main inferences regarding the results shown are relating to trade-offs, it is necessary to discuss each trait generation by generation. This is in part relating to the theory of trade-offs between survival and reproduction which is central to several theories of ageing and especially

important when discussing lifespan extension under DR. We have however taken the many comments suggested by yourself, Reviewer 2 and the associate editor on board, with the idea of making the results and methods section easier to read.

In sum, I feel that this is a potentially interesting study reporting novel findings but the clarity of presentation could be much improved - this would make the paper much more readable and thus more impactful.

Minor/detailed comments:

Intro: the intro is generally clearly and well written; the questions are well laid out.

Thanks!

L39: temporary vs. intermittent fasting: difference?

These terms are slightly different yet still quite similar. In order to avoid confusion, we have added a slight extension to the sentence to explain the difference (Lines 40-44).

L68: How does larval starvation differ from DR? This may not be clear to the generalist reader; in many animals starvation decreases lifespan, unlike DR, so some clarification may be necessary here. **This is an important point.). We have expanded upon this in the discussion on lines 74-75.**

Methods: in general, presentation/explanation could be improved; design rather complex and needs better explanation.

We appreciate the various comments about improving the methods. We hope to have addressed each of the suggestions and believe this section is far easier to read.

L121, and elsewhere: 4 plates of 4 different types of plates? Not clear.

We have made this explicit on Line 133.

L121, and elsewhere: regarding the 4 treatments, how much replication was there for each of these treatments?

This is now detailed in Figure 1A.

L121, and elsewhere. Similarly, how were the experimental blocks defined? Maybe indicate those, or at least mention them, also in Fig.S1?

This is mentioned on Line 187.

I agree that a figure (or two, if necessary), in the spirit of Fig.S1, is the perhaps best way to convey the details of the experimental design. However, Fig.S1 does not do this job adequately (yet): there is lots of relevant information about the design missing from the legend of that figure. The legend of that figure should be expanded to include more detail of what is actually shown in the figure (e.g., the legends say “discussed below” – but I could not find any discussion in the supplementary file). Replication and blocks etc. should also be clarified here. The figure explaining the experimental design is quite central for understanding the experiments – maybe consider moving the figure into the main manuscript?

We have since readded this into the methods section with increased detail.

L141: Might this not lead to potential confounding effects? In some ageing studies (in rodents, etc.) it has been found that variation in “birth date” can have a significant impact on adult lifespan but is often unaccounted for in statistical models. Maybe a caveat should be included here?

This is a good point. There is a lack of evidence for the Lansing effect in this species (older mothers giving birth to shorter lived progeny, Travers, 2021 (Beneficial cumulative effects of old parental age on offspring fitness). In fact, increasing age leads to larger eggs which causes an increase in reproduction in the larger progeny. However, as the eggs produced from DR parents are in fact smaller than AL despite being produced by older mothers, and we still see an increase when parent-offspring DR match, we can be confident that this is in fact a true result and not a result of confounding effects.

L149-151. This explanation / rationale is insufficiently clear to me.

The predominant reason was for sample size purposes as the number of individuals per treatment and number of treatments was not pragmatic if all combinations were used.

L161. Blocks. Maybe use a figure to clarify the design? Also see comments above.

We have readded in Figure 1 which outlines the experimental design.

L172-174. This is also a bit hard to understand – why was this done? What's the underlying rationale / reason?

We agree that this could have done with a bit more explanation. We have since added some more information on Lines 201-204.

L182. Incomplete sentence – reword.

Done.

L186-L190, and elsewhere. It might be best to give the actual statistical models as equations – this would be clearer and more explicit.

This is a good point, we have made this explicit in Table S18.

L186-L190. Issue replication mentioned further above; definition of blocks, etc. – as mentioned above, the experimental design as described in the M&M section is not sufficiently clear. A figure and an improved written description of the design would be important for the reader to grasp the details of the design.

We have added figures 1A and 1B and hope to have expanded upon the methods section sufficiently.

L130. Similar or the same fixed and random effects? Vague. Perhaps better give the model equation?

These are now given in Table S18.

Fig.1A-D legend. It should read "Cox" not "cox".

Changed.

L405-406. Why is there this difference? The reader is being told this fact but the generalist reader who does not know that previous paper might have a hard time understanding what this finding means vis-à-vis the findings reported here. Is this a discrepancy or not? If it is, how could it be explained? See above: how is larval starvation in *C. elegans* different (or not) from DR?

This is an important point and we have expanded upon this in the discussion on lines 460-462.

Referee: 2

Comments to the Author(s)

Dietary restriction (DR) features prominently in the biology of aging. It has often been shown that organisms fed a restricted diet live longer than those fed an unrestricted diet, but that there are trade-offs with fecundity. One variety of dietary restriction involves temporary fasting (TF), which has clinical applications in humans. The effects of DR have been shown to carry over to the offspring and grand-offspring of individuals subjected to DR in a variety of organisms ("cross-generational" effects), and sometimes to the F3 and beyond ("transgenerational" effects). Various proximate (physiological) and ultimate (evolutionary) models have been posited to explain the phenomenon of DR-enhanced lifespan, with its (apparent) concomitant tradeoffs in fecundity.

The authors report a study using a study with the nematode model *C. elegans*, in which they investigate several questions concerning the cross-generational and transgenerational effects of TF on lifespan, fitness, and growth. Specifically, they address the questions (their words): "...addressing the following unresolved questions: 1) How does TF affect mortality risk and reproductive ageing once the animals return to their standard food regime?; 2) How do offspring of fasting parents perform in matching and mis-matching environments?; 3) Do transgenerational effects of ancestral fasting shape mortality risk and reproductive ageing of distant descendants?; and 4) Does reduced reproduction under DR represent a decision-making strategy?"

Worms are assigned to one of four dietary treatment groups: Ad libitum (AL), food odor (FO), AL+FO, and Temporary Fasting (TF). Individual lineages were carried out to the F3 generation (sometimes

less, depending on the question of interest). Phenotypes of interest are lifespan, lifetime fecundity, individual fitness (the dominant eigenvalue of the Leslie matrix derived from the life-table), and body size measured at various life stages. Individual lineages can be assigned a three or four generation dietary state using the notation (P0-F1-F2) or (P0-F1-F2-F3), where the state at each generation is one of the four treatment groups (e.g., in the FO-TF-AL group, the parent was raised under the food odor treatment, the F1 in the temporary fasting treatment, and the F2 in the AL treatment). From this design, the generation-specific effects of treatment can be quantified, and the fitness effects assessed.

This a very elegant experiment, and the results are admirably transparent. I think the interpretation is largely sensible.

Thanks!

I do have a quibble with the authors' terminology, which is that they use the terms "positive" and "beneficial" seemingly interchangeably, and likewise the terms "negative" and "deleterious". "Positive" and "negative" are straightforwardly interpreted as an increase or decrease in trait value, even when the trait value is the dominant eigenvalue of the Leslie matrix ("individual fitness", which is a sensible definition). However, "beneficial" implies "favored by natural selection" (and similarly "deleterious"=disfavored), and in the absence of theoretical analysis (or a selection experiment), it's impossible to say what would be favored by natural selection.

This is a valid point and we have made the relevant changes in text throughout the manuscript

However, I think that the authors' have shown pretty clearly that cross-generational effects that are unambiguously positive may be negative in further generations in particular circumstances.

I have one non-trivial criticism, which concerns the presentation of the analysis. The model(s) needs to be written out explicitly, preferably in math or at the very least in pseudo-code. The interested reader should be able to re-create the analysis, and unless one is familiar with the specific R packages in question, that does not seem possible (and probably not even then, I bet).

We agree that this would help the reader and have now included a table detailing the various models (Table S18).

Also, Figure S1 needs to be Figure 1 in the main text. I could not make it through the results without constantly referring to the figure.

We agree, this was originally moved to the supplement. It has now been replaced into the MS as Figure 1.

Some minor comments, by Page/line.

P8/L149. Should be Figure S1 (?)

Done

P12/L247 (and throughout). The authors have a tendency to use syntax that is the opposite of what is shown in a figure. Here, they say "...offspring show decreased mortality when placed in AL conditions". Which is true, but the figure shows the AL treatments on the positive side. If you say "increased survivorship", the human brain (or at least my brain) has an easier time of it.

We have since rewritten some of the results and have addressed this throughout.

P15/L300. I like the (P0-F1-F2) notation, but it could be interpreted the other way around, and the reader has to do more thinking than necessary. Write out the interpretation (i.e., just like I wrote it) the first time you use it.

This is a good point and we have made this explicit at the start of both the F2 and F3 sections.

P15/L314. "...TF grandparents with AL parents (FO-TF-TF:...)" I think should be ...(FO-AL-TF).

This is correct, we have changed this.

P15/L316. from the sentence, should this be reversed, i.e., AL-AL-TF? But you reported AL-AL-TF above (line 312). Something is not right here.

This was incorrect and we have now since replaced the sentence.

Appendix B

Response to Reviewers

Associate Editor

Comments to Author:

I obtained one review for this resubmitted version as well as having read it carefully myself. I agree with the reviewer that the revision reads better and the results are interpreted in a more interesting way in the Discussion. I also agree with reviewer 1 that the odds plots should be replaced by survival curves; the odds ratios are anyway given in the text.

Thank you, this is a good suggestion to add survival curves. We produced survival plots as suggested, however, we prefer to have them in the supplementary material rather than in the main text (they are now correctly referenced as mentioned by the reviewer).

There are two reasons for this. First, the forest plots help visualise our analyses, while survival curves do not, Humans are bad at visualising the numbers, so it helps to show the effect size and the direction of the effect. Second, because we have many treatments to compare, the survival curves do not allow sufficient visual clarity. We now have both forest plots that visualise our analyses and survival plots as well.

I have one fairly substantial issue that still needs to be addresses. I did not raise it in the previous round of reviews because I only noticed the relevant piece of information at this reading. I apologize for what may seem like moving goalposts, but I feel partly excused because this information is rather buried in the Methods. Namely, the authors mentions "disproportionate day one and two mortality of TF individuals" (l. 154). They bring this up in the context of the sample sizes, but I think this – apparently unusually high – mortality under the temporary fasting treatment has important implications for interpretation of the results.

First, it seem to contradict what is said in l. 439ff about the TF treatment not resulting in malnutrition. If the TF did not cause malnutrition, why were these young worms dying?

Second, this opens the possibility that this mortality might have been non-random and thus may have imposed some selection, the consequences of which might have contributed to the results. Could some of the trans-generational effects be due to genetically- based response to selection? How genetically variable was the strain used? Even if the strain was isogenic (info on genetic variation of the strain is not provided), could it be that TF preferentially eliminated phenotypically frail individuals that would otherwise be first to die of old age, thus biasing the lifespan upwards?

The paper needs be more upfront with this issue. The information about the magnitude of this mortality during the TF and other treatments should be provided and the issue of potential selection confound brought up in the Discussion. I the authors have arguments as to why this mortality is unlikely to have biased the results, these arguments should be laid out in Discussion; otherwise, the caveats should be made explicit.

Sorry if this was unclear. TF, as well as any other form of DR, causes *C. elegans* to search for food on plates which results in accidental "walling". "Walling" means that worms crawl on the side of the plate and desiccate and die. This is very common in all such experiments and such individuals are universally censored.

The strain that we used is isogenic and accidental walling does not affect the transgenerational effects. We have no reason to believe that it affects lifespan in any way either.

We changed the phrasing in the methods (Line 154) to make it clear that we refer to accidental walling and not death from malnutrition.

Except for this point, I have a few minor editorial suggestions:

l. 28: "in the parental generation": to minimize risk of confusion, I would rephrase "of the individuals subject to TF" ("parental" refers to parents to some focal individuals, which are not clear here), and

then I would replace "future generations" by "their descendants" (as only descendants of the treated individuals are affected and not the entire future generations). Surely, most evolutionary biologists would have no problem with understanding the current version correctly, but abstracts can be read by a wide audience, so it is good to be very precise.

This is a good point – we have changed this in text.

I. 88-89: it is not clear from the sentence structure if the clause "suggesting that..." refers to the effect of DR or the effect of food odor reversing the effects of DR.

We have changed this sentence on lines 87-89 to make this distinction clearer.

I. 96 "distant" => "more distant" or "descendants beyond offspring"?

Changed.

I. 97: I am not convinced that the design would have allowed to discern whether or not this is a "decision-making strategy", in particular in the absence of any operational definition of "strategy". It would be better to rephrase this question in terms closer to what is actually shown, i.e. , response mediated by odor perception (or absence thereof).

This is a good suggestion and we have added your example.

I. 470: "integral to this increase" : meaning what exactly?

This increase was originally referring to increased mortality. As such, we have changed "increase" to "decrease" to match the previous sentence.

Reviewer(s)' Comments to Author:

Referee: 2

Comments to the Author(s).

The authors have satisfactorily addressed the comments I provided in my previous review. The other reviewer noted that the authors should present a figure with the survivorship curves, which was a good idea. Those figures (S4) are not referenced in the paper. In fact, I think the survivorship curves should replace the line diagrams of the odds ratios in Fig. 2, which don't add anything beyond the values reported in the text.

Thank you for this comment. We have answered regarding the survival curves above and have correctly referenced them in the paper.